# Ultra-low extracorporeal volume microfluidic leukapheresis is safe and effective in a rat model

Mubasher Iqbal [1], Alexandra L. McLennan [2,3], Anton Mukhamedshin [1], Mai T. P. Dinh [1], Qisheng Liu[3,4], Jacob J. Junco [2], Arvind Mohan[2], Poyyapakkam R. Srivaths [2], Karen R. Rabin[2], Thomas P. Fogarty III [2], Sean C. Gifford[5], Sergey S. Shevkoplyas [1] ✉ & Fong W. Lam [2,3] ✉

Leukapheresis is a potentially life-saving therapy for children with symptomatic hyperleukocytosis. However, the standard centrifugation-based approach exposes pediatric patients to significant complications due to its large extracorporeal volume, high flow rates, and considerable platelet loss. Here, we tested whether performing cell separation with a high-throughput microfluidic technology could alleviate these limitations. In vitro, our microfluidic devices removed ~85% of large leukocytes and ~90% of spiked leukemic blasts from undiluted human whole blood, while minimizing platelet losses. Multiplexed devices connected in parallel allowed for faster, clinically relevant flow rates in vitro with no difference in leukocyte collection efficiency. When connected to Sprague-Dawley rats, the devices removed large leukocytes with ~80% collection efficiency, reducing the leukocyte count in recirculating blood by nearly half after a 3-hour procedure. Evaluation of plasma biomarkers and end-organ histology revealed no adverse effects compared to sham control. Overall, our study suggests that microfluidics-based leukapheresis is safe and effective at selectively removing leukocytes from circulation, with separation performance sufficiently high to ultimately enable low extracorporeal volume leukapheresis in children.

Leukemias are the most common type of cancer in children, with an annual incidence of ~5 per 100,000 children in the United States[1]. Up to 20–30% of patients with acute leukemia develop hyperleukocytosis, defined as a circulating white blood cell (WBC) count greater than 100,000 per μL of predominantly leukemic blasts[2], placing patients at risk for life-threatening complications like tumor lysis syndrome, disseminated intravascular coagulation, and end-organ hypoperfusion from symptomatic leukostasis[3–6].

Although definitive treatment for acute leukemia involves chemotherapy, therapeutic leukapheresis – an extracorporeal procedure for removing leukemic blasts from recirculating blood[7] – may be indicated in patients with symptomatic hyperleukocytosis (leukostasis) to reduce the burden of leukemic blasts faster than could be achieved following initiation of chemotherapy[8,9]. However, current centrifugation-based leukapheresis machines have significant limitations, including large (hundreds of mL) extracorporeal volumes (ECV)

[1]Department of Biomedical Engineering; University of Houston, Houston, TX, USA. [2]Department of Pediatrics; Baylor College of Medicine, Houston, TX, USA. [3]Center for Translational Research on Inflammatory Diseases; Michael E. DeBakey Veterans Affairs Medical Center, Houston, TX, USA. [4]Department of Medicine; Baylor College of Medicine, Houston, TX, USA. [5]Halcyon Biomedical Incorporated, Friendswood, TX, USA. ✉e-mail: sshevkop@central.uh.edu; flam@bcm.edu

and high flow rates (tens of mL min$^{-1}$) which are required to establish a stable separation interface during blood centrifugation[10]. These limitations are compounded in children by their relatively small estimated total blood volumes (eTBV), which may necessitate extracorporeal circuits primed with red blood cells (RBC) to avoid dilutional anemia, placement of large central catheters (CVL), and sedation during the procedure[11–15]. In the smallest patients, centrifugation-based leukapheresis may not be possible due to the above reasons[14].

Replacing centrifugation with microfluidic cell separation may obviate these limitations because microfluidic devices typically have vanishingly small dead volumes and generally do not require high flow rates to maintain separation efficiency[16–20]. As a result of recent advancements, several microfluidic technologies have demonstrated highly efficient separation of WBC from whole blood (WB); however, technical challenges reduce their feasibility for leukapheresis[16]. For example, micro-filtration devices suffer from low throughputs and clogging[21,22], inertial microfluidic devices require significant dilution[23,24], and devices based on deterministic lateral displacement require sheath fluid and precise flow control[25,26].

However, microfluidic devices based on controlled incremental filtration (CIF)[27] appear promising because they can separate WBC efficiently from a variety of blood sources[28–30], can be driven by various pumping methods[29,31,32], and have scalable throughputs from tens of μL min$^{-1}$ to tens of mL min$^{-1}$ through multiplexing without losing separation efficiency[29,32]. We previously reported that a CIF device designed for diluted blood (10% hematocrit) can remove WBC with >85% collection efficiency in vitro; however, the design's separation performance was severely diminished at higher hematocrits, including undiluted whole blood[30]. Therefore, important challenges remain unresolved, such as if the CIF technology would work with undiluted WB at clinically relevant flow rates, if standard clinical anticoagulation strategies would be sufficient for extended periods in vivo, and if repeated passages through a CIF device would lead to adverse effects on the blood cells or patients[27,31,32].

Therefore, the primary objective of this study was to test the safety and efficiency of WBC removal from undiluted WB within a living animal using a CIF-based microfluidic device operating in the recirculation regime (microfluidic leukapheresis). The secondary objective was to test whether CIF could remove leukemic blasts from undiluted human WB with collection efficiency meeting or exceeding performance metrics for centrifugal leukapheresis observed in our clinical practice and in the literature. We hypothesized that CIF-based microfluidic leukapheresis would be well-tolerated by animals, would preferentially deplete WBC while minimizing RBC and platelet (PLT) loss in vivo with minimal cytolysis, and would remove leukemic blasts from human blood efficiently in vitro. Our results suggest that microfluidic leukapheresis enabled by CIF is safe and effective in animals and, with further development, could potentially fulfill the unmet need for low-ECV, less invasive therapeutic leukapheresis in pediatric patients.

## Results

### Current pediatric leukapheresis processes

To quantify the real-world effectiveness of centrifugal leukapheresis in reducing WBC counts and understand the impact of its limitations on patient care, we assessed 43 pediatric leukapheresis procedures performed for hyperleukocytosis in 37 patients at a large, urban, quaternary care pediatric hospital. The median age and weight were 11 years and 46 kg with most patients being male and Hispanic/Latinx (Fig. 1a). Most patients had acute lymphoblastic leukemia (79%; Fig. 1b) and required only one leukapheresis session during their admission (88%), with 53% and 37% having symptomatic hyperleukocytosis or tumor lysis syndrome, respectively, at the time of leukapheresis (Fig. 1c). Patients were admitted to the pediatric intensive care unit (PICU) for CVL placement and leukapheresis using either the COBE® Spectra or Spectra Optia® systems. The median times to leukapheresis from emergency department (ED) arrival, PICU admission, and CVL insertion were 968, 522, and 131 min, respectively, with the median duration of the leukapheresis procedure being 281 min (Fig. 1d). The median time to protocol-based chemotherapy after the initial PICU admission was 26 h [19–38 h], like in previous reports[33].

The ECV of the leukapheresis circuits was ~300 mL (representing ~9% [7–13%] of the patients' eTBV) and the median inlet flow rate was 44.9 mL min$^{-1}$ [29.2–60.0 mL min$^{-1}$]. Consequently, one-third of patients required RBC priming of the circuit. Most patients were transfused within 2 h of leukapheresis (excluding RBC priming), with 49% receiving RBC and 37% receiving PLT (Fig. 1e). Most procedures in which PLT transfusions were given (15/16) occurred during or after leukapheresis. To prepare for leukapheresis, >90% of patients received sedatives (in addition to analgesics) with one-third requiring endotracheal intubation and mechanical ventilation solely for CVL placement (Fig. 1f). There was an overall decrease in blood pressure (BP) during leukapheresis (Fig. 1g–i), although only 16% met Pediatric Advanced Life Support criteria for hypotension[34] and 19% required initiation or increase in inotropic or vasoactive support (Fig. 1f). Patients also suffered from hypocalcemia (28%; Fig. 1f). Overall, ~90% of patients required additional resources or circuit priming with RBC that could be directly attributed to the large ECV, the need for large-bore catheters, and/or high flow rates associated with centrifugal leukapheresis.

All patients had hyperleukocytosis at the time of leukapheresis initiation (baseline) and experienced significant reductions of WBC (50% [38–65%]) and PLT (32% [18–48%]) counts throughout the procedure, which were also significantly lower than baseline at their nadir during leukapheresis (Fig. 1j, k). To achieve such reductions, a median of 3.6 [2.7–4.6] eTBV was processed during each leukapheresis procedure, which was equivalent to a processing rate of 0.8 [0.6–1.0] eTBV per hour (Fig. 1l).

### In vitro testing of microfluidic devices using undiluted human whole blood

Because leukemic blasts are typically as large as the largest normal leukocytes[35], we based the microfluidic device design on our previous experience with developing CIF technology for separating large WBC (granulocytes and monocytes) from smaller blood cells (lymphocytes, RBC, and PLT)[32]. The basic element of a CIF device consists of three microfluidic channels positioned side by side and separated from each other by two arrays of identical (~20 μm-wide) filtration gaps (Fig. 2a). In such a design, a small fraction (~10$^{-3}$) of the fluid flow in the middle channel (retentate) is extracted through each filtration gap into the side channels (filtrate). The width of the extracted flow lamina $w_l$ determines the size threshold for the cells that are small enough to be pulled into the filtrate channels. Consequently, cells larger than the size threshold (i.e., large normal WBC and leukemic blasts) preferentially accumulate in the retentate channel (Fig. 2a). When designing a CIF element, the widths of the retentate and filtrate channels are gradually adjusted (using a previously described iterative framework[27,28]) to accommodate the outflow of fluid from the retentate channel and to keep the width of the extracted flow lamina constant along the length of the device. The amount of fluid extracted through each gap and the total number of gaps in the design determine how much of the initial sample volume is ultimately collected as filtrate relative to retentate (i.e., device flow ratio)[28,32].

In this study, each microfluidic device consisted of eight separate CIF elements of the same design connected in parallel to allow processing of the sample at a flow rate of 1.2 mL min$^{-1}$ (Fig. 2b, Supplementary Fig. 1). Driven by a syringe pump, blood entered the device through a common input port and was distributed to the CIF elements via a network of bifurcating channels. The output of all filtrate channels was collected via a network of converging channels and extracted from

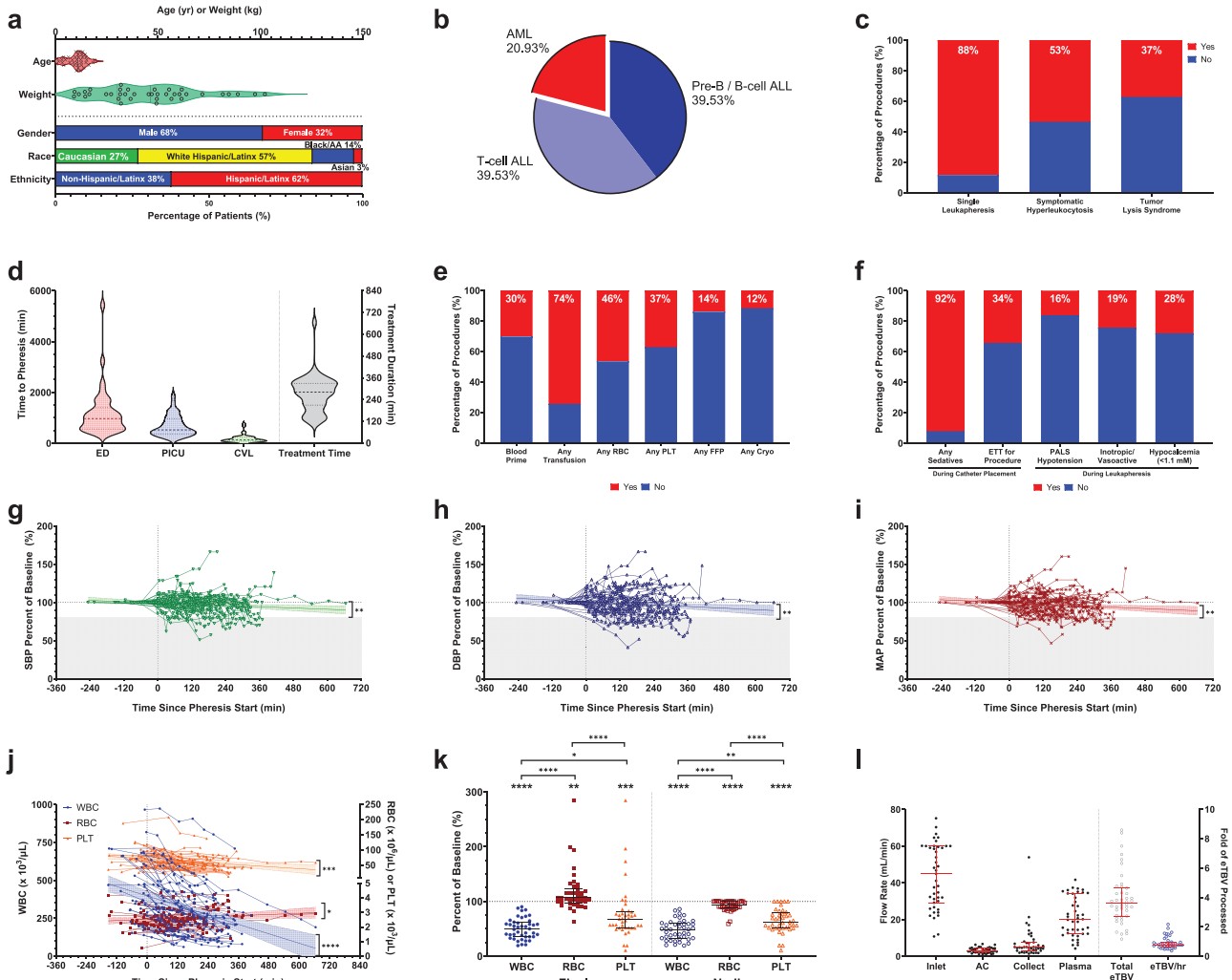

**Fig. 1 | Demographics, complete blood count trends, and clinical outcomes of children receiving centrifugation leukapheresis for acute leukemia. a** Age, weight, and demographics as well as (**b**) leukemia types in patients receiving centrifugation leukapheresis. **c** Percentage of procedures ($n = 43$) that were single procedures, performed for symptomatic hyperleukocytosis, or had documented tumor lysis syndrome. **d** Times to leukapheresis from ED arrival, PICU admission, or CVL insertion (left) and leukapheresis treatment duration (right). **e** Blood product exposure within 2 h of leukapheresis. **f** Resource utilization or complications in preparation of or during leukapheresis. **g**–**i** As compared to baseline values, there was an overall decrease in systolic BP (SBP; $p = 0.0065$), diastolic BP (DBP; $p = 0.007$), and mean arterial pressure (MAP; $p = 0.003$) throughout the leukapheresis procedure (shaded regions signify a > 20% decrease from baseline blood pressures). **j** WBC (●; left axis; $p < 0.0001$) and PLT (▲; right axis; $p = 0.0003$)

significantly decreased while RBC (■; right axis; $p = 0.0497$) increased over time during leukapheresis. **k** As a percentage of baseline cell counts, the final WBC ($p < 0.0001$) and PLT ($p = 0.0002$) counts were significantly decreased and RBC ($p = 0.0026$) was slightly increased, whereas nadir WBC, PLT, and RBC were all ($p < 0.0001$) significantly decreased as compared to baseline. **l** Inlet, anticoagulation (AC), collection, and plasma flow rates (left y-axis) and the total eTBV and eTBV per hr processed (right y-axis) for each procedure. Data shown as median [IQR]. Data analyzed by (**g**–**j**) linear regression with slope and 95% CI shown and (**k**) Kruskal-Wallis test with Dunn's multiple comparison test comparing cell types to each other and one sample Wilcoxon test with Pratt correction comparing each column to an expected median value of 100 (signifying baseline percentage). *$p < 0.05$, **$p < 0.01$, ***$p < 0.001$, ****$p < 0.0001$.

a common outlet port as the filtrate. Because the middle channel ports (retentate) in each CIF element were surrounded by side channels (filtrate; see Fig. 2a and Supplementary Fig. 1a, b), we used a separate manifold layer to collect the retentate from each CIF element and direct it to a single outlet (Fig. 2b, c).

We first tested whether a CIF-based microfluidic device could effectively separate the largest normal WBC (for human, monocytes [MON] and eosinophils [EOS])[36] from undiluted WB. We generated four different designs of the same overall architecture (Fig. 2a–c, Supplementary Fig. 1) but with different flow fractions extracted through the filtration gaps (and, therefore, different cell size thresholds)[28,32]. All the CIF designs increased the concentration of every WBC subtype in the retentate (and decreased it in the filtrate) (Supplementary Fig. 2a, d–h). None of the designs decreased the RBC and PLT concentrations in the

filtrate with respect to the input samples (Supplementary Fig. 2b, c). The collection efficiency (CE; the percentage of cells passing through the device that were collected in the retentate) depended strongly on the cell type and was the highest for the largest WBC (MON and EOS) and the lowest for the smallest blood cells (RBC and PLT) for each design (Fig. 2d, e). Across all CIF designs, CE for EOS ranged from $75.9 \pm 7.5\%$ to $86.9 \pm 5.1\%$, and for MON from $72.9 \pm 3.8\%$ to $85.0 \pm 3.0\%$. CE for RBC (signifying a loss) was similar for all designs ($5.2 \pm 0.2\%$), approximating the expected value (~4.5%) based on the device flow ratio (ratio of the filtrate to retentate volume), which was $21.1 \pm 0.6$[30,32]. On average, PLT CE ranged from $7.8 \pm 0.5\%$ to $11.6 \pm 0.9\%$, or about twice the expected value.

The ability of the CIF devices to concentrate and remove large WBC is further highlighted by the drastic difference between the

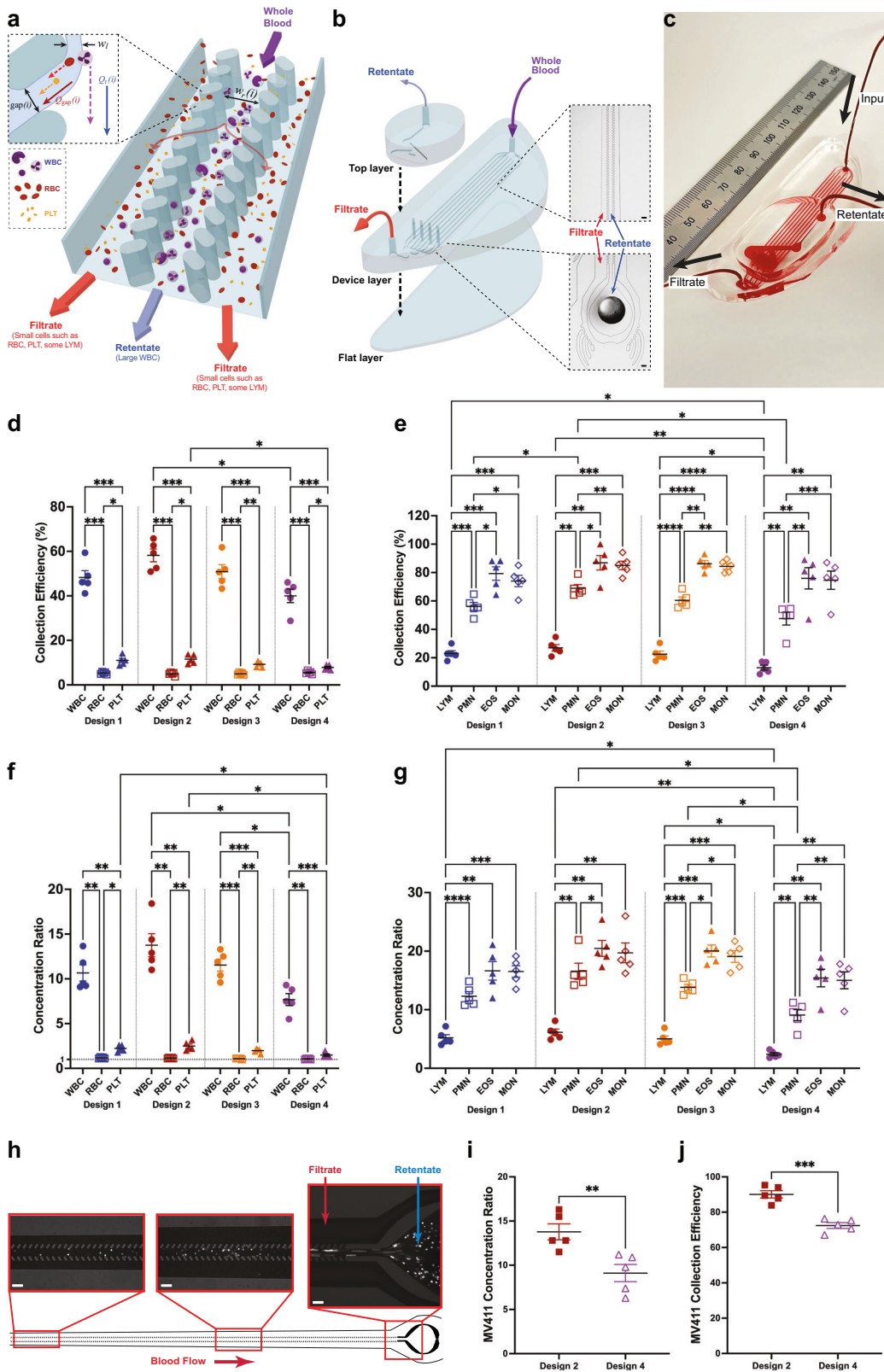

concentration ratio (CR; retentate count divided by input count) for EOS (20.3 ± 1.4) and MON (19.9 ± 1.6), and small concentration ratios for RBC (1.1 ± 0.0) and PLT (2.0 ± 0.2) (Device 2; Fig. 2f, g).

To determine the CIF devices' efficacy in removing leukemic blasts, we spiked undiluted human WB samples with fluorescently labeled MV-4-11 cells (Fig. 2h, Supplementary Movies 1) and passed them through CIF devices with the highest (Design 2) and the lowest

(Design 4) CE for EOS. Both designs concentrated the spiked MV-4-11 cells in the retentate significantly with a CR of 13.8 ± 0.9 for Design 2 and 9.1 ± 1.0 for Design 4 (Fig. 2i), and consequently removed most of MV-4-11 leukemic blasts from the spiked WB samples with a CE of 90.1 ± 2.1% for Design 2 and 72.5 ± 3.7% for Design 4 (Fig. 2j, Supplementary Fig. 2i).

**Fig. 2 | Schematics and CBC measurements from in vitro, single-pass experiments from healthy donor whole blood and healthy donor whole blood spiked with leukemic cells. a** Schematic showing how the CIF channels operate. As whole blood enters a CIF element, smaller cells (e.g., RBC and PLT) exit through the gaps $i$, while larger cells (e.g., WBC) remain in the central retentate channel. **b** Schematic showing the assembly of an 8-element CIF device. As whole blood enters the input port of the device layer (top right), the blood is separated into 8 parallel CIF elements. Each element's filtrate is merged and exits from the filtrate output port. WBC in the retentate exit vertically into a collecting top layer and are removed from the retentate output port. **c** Photograph of a blood-filled CIF device (ruler shown in mm). **d**, **e** Collection efficiency data highlights CIF's ability to separate large cells. RBC and PLT loss are kept to <15%, while large WBCs such as MON and EOS are collected up to an efficiency of ~85%. **f** The concentration ratio of all designs was significantly higher for WBC than RBC and PLT, with the RBC concentration ratio being close to 1 across all designs. **g** Among WBC subtypes, EOS and MON exhibited the highest concentration ratios across all designs, followed by PMN. LYM, due to their small size, were least concentrated. **h** Images of fluorescently labeled MV-4–11 cells spiked into whole blood flowing within the device. Note how most leukemic cells are located within the retentate channel with little to none in the filtrate channels. **i**, **j** Both designs were able to effectively concentrate (**i**, $p = 0.0056$) and remove (**j**, $p = 0.0007$) MV-4–11 leukemic cells from whole blood, with Design 2 being more efficient than Design 4. Data shown as mean ± s.e.m. $n = 5$ for both healthy and MV-4–11 spiked blood data. Scale bar = 100 μm; $w_l$=width of the fluid lamina, $Q_r(i)$=retentate channel flow, $Q_{gap}(i)$=filtration gap flow. Data analyzed by (**d**–**g**) 2-way RM ANOVA with Tukey's multiple comparison test or (**i**, **j**) 2-tailed paired t-test. *$p < 0.05$, **$p < 0.01$, ***$p < 0.001$, ****$p < 0.0001$.

To ascertain CIF's ability to operate at clinically relevant throughputs without losing performance, we parallelized multiple Design 2 (highest CE for MON and EOS) devices via Y-connectors (Fig. 3a, b; Supplementary Fig. 4). One, two, four, and eight CIF devices connected in parallel were perfused with undiluted WB at 1.2, 2.4, 4.8, and 9.6 mL min$^{-1}$, respectively. There were no differences in CE or CR for any cell type across all flow rates. The WBC, RBC, and PLT CE were 56.7 ± 0.5%, 5.8 ± 0%, and 9.0 ± 0.2%, respectively, with the CR of WBC being significantly higher than RBC and PLT (Fig. 3c, e). The WBC subtypes were also collected similarly among all flow rates, with MON and EOS CE being 84.3 ± 0.5% and 81.6 ± 1.4%, respectively, and MON CR (17.6 ± 0.2) being the highest among all groups (Fig. 3d, f).

### Efficacy of microfluidic leukapheresis in rats
To evaluate the safety and efficacy of CIF-based microfluidic leukapheresis, we performed a 3 h leukapheresis procedure in anesthetized Sprague-Dawley rats. The animals were given intravenous heparin (500 U kg$^{-1}$)[37–39] for anticoagulation prior to connecting a microfluidic device (CIF group) or a resistance bridge (sham group) between the carotid artery and jugular vein. The arteriovenous pressure difference was sufficient to drive the flow through the extracorporeal circuit without a perfusion pump (Fig. 4a, b). Four male and four female rats of similar weights (CIF: 375 ± 28 g; sham: 267 ± 22 g; $p = 0.82$) were used in each group. The ECV of the CIF circuit was ~0.74 mL (mostly from the tubing connecting the device to the animal), accounting for only 2.9 ± 0.2% of the animals' eTBV.

Baseline cell counts were similar between the CIF and sham groups (Supplementary Fig. 5a). Although animals in the CIF group had lower overall BP, the trends in mean arterial pressure (MAP) were similar and remained at or above the starting MAP throughout the procedure for both groups (Fig. 4c). Likewise, the mean total flow rate remained stable throughout the experimental duration (Fig. 4c). While the overall mean flow rate was higher in the CIF compared to the sham group (0.448 ± 0.032 vs. 0.319 ± 0.043 mL min$^{-1}$, respectively; $p < 0.05$), the number of eTBV processed was similar between the two groups (Fig. 4d). Additionally, the volume (3.3 ± 0.4 eTBV) and rate (1.1 ± 0.1 eTBV per hour) of blood processed via CIF-based leukapheresis in rats mirrored our pediatric patient cohort (Fig. 1i). Because we continuously removed retentate during leukapheresis, animals in the CIF group had more fluid removed and given than those in the sham group (Fig. 4e).

Rats in the CIF group had a significantly large decrease in circulating WBC, with a small decrease in final RBC and no difference in PLT as compared to sham-treated animals (Fig. 4f–h). As expected, CE depended strongly on cell type and was the highest for EOS (81.5 ± 28.3%) and neutrophils (PMN; 74.3 ± 20.2%), the two largest WBC subtypes in rats[36]. CE was the lowest for RBC (5.5 ± 1.3%) and PLT (4.3 ± 1.1%; Fig. 4k), in agreement with the theoretical value (~4.2%) expected based on the average device flow ratio (22.6 ± 1.9)[32]. To further quantify the efficiency of WBC separation, we calculated the CR by dividing cell counts in the serially collected retentate samples by the contemporaneous values in recirculating blood. Like in our in vitro experiments with human blood, the CIF device was highly effective at preferentially removing the largest WBC from the circulation, as compared to smaller WBC subsets, RBC, and PLT (Fig. 4j). As a result, by the end of the leukapheresis procedure, animals in the CIF group experienced a significant reduction in their circulating WBC count from baseline (45.6 ± 2.9%), with a smaller reduction in RBC count (19.4 ± 2.4%) and no difference in PLT count (Fig. 4f–h, Supplementary Table 1). This reduction in overall WBC count was associated with significant decreases in counts for the major WBC subsets (Fig. 4i, Supplementary Table 1). Rats in the sham group had no reduction in their WBC count, as expected. However, their RBC count decreased by 13.2 ± 1.3% (likely due to the serial sampling) and their PMN count significantly increased (likely due to demargination caused by surgical stress[40,41]).

We found no relationship between the final cell counts expressed as a percentage of their baseline (WBC$_{final\%}$, RBC$_{final\%}$, and PLT$_{final\%}$) and the amount of fluid removed or given (Supplementary Fig. 5b). Using multiple linear regression analysis, we found that being in the CIF group was independently associated with WBC (but not RBC or PLT) removal. Furthermore, the amount of fluid removed/given was not independently associated with WBC$_{final\%}$ (Supplementary Table 2). To determine whether the changes in cell counts were due solely to serial blood sampling, we calculated a relative ratio comparing PLT$_{final\%}$ and WBC$_{final\%}$, using changes in RBC$_{final\%}$ to normalize the values. A ratio of <1 signifies greater loss of a cell type as compared to RBC; conversely, a ratio of >1 signifies greater RBC loss as compared to the other cell type. As compared to the sham group, rats in the CIF group had similar PLT$_{final\%}$:RBC$_{final\%}$, but significantly lower WBC$_{final\%}$:RBC$_{final\%}$ (Fig. 4l). Because WBC loss exceeded RBC loss in the CIF group, these data suggest that WBC were preferentially removed by the CIF device.

### Safety of the CIF-based microfluidic leukapheresis in rats
To assess whether the CIF-based microfluidic leukapheresis caused end-organ injury, we collected plasma before and after each procedure. In a subset of animals, hematoxylin and eosin (H&E) stained tissue sections of the lung, kidney, and liver were examined to assess for gross changes in histology between CIF and sham-treated animals. When comparing CIF to sham animals, there were no differences in plasma markers of organ injury or hemolysis before or after the experiments (Fig. 5). In both groups, there was an increase in blood urea nitrogen and creatinine (Fig. 5d, e) as well as a small, but clinically insignificant, increase in potassium (Fig. 5k). In the sham group only, there was a small decrease of lactate (Fig. 5b) and an increase of lactate dehydrogenase (Fig. 5j). There were no gross differences between the groups on H&E or Wright stains (Fig. 5c, f, i, l, Supplementary Fig. 6). In both groups, there was a rise in plasma von Willebrand factor (VWF) and thrombin-antithrombin (TAT) complexes after the procedure (Fig. 5m, o). However, fibrinogen levels in

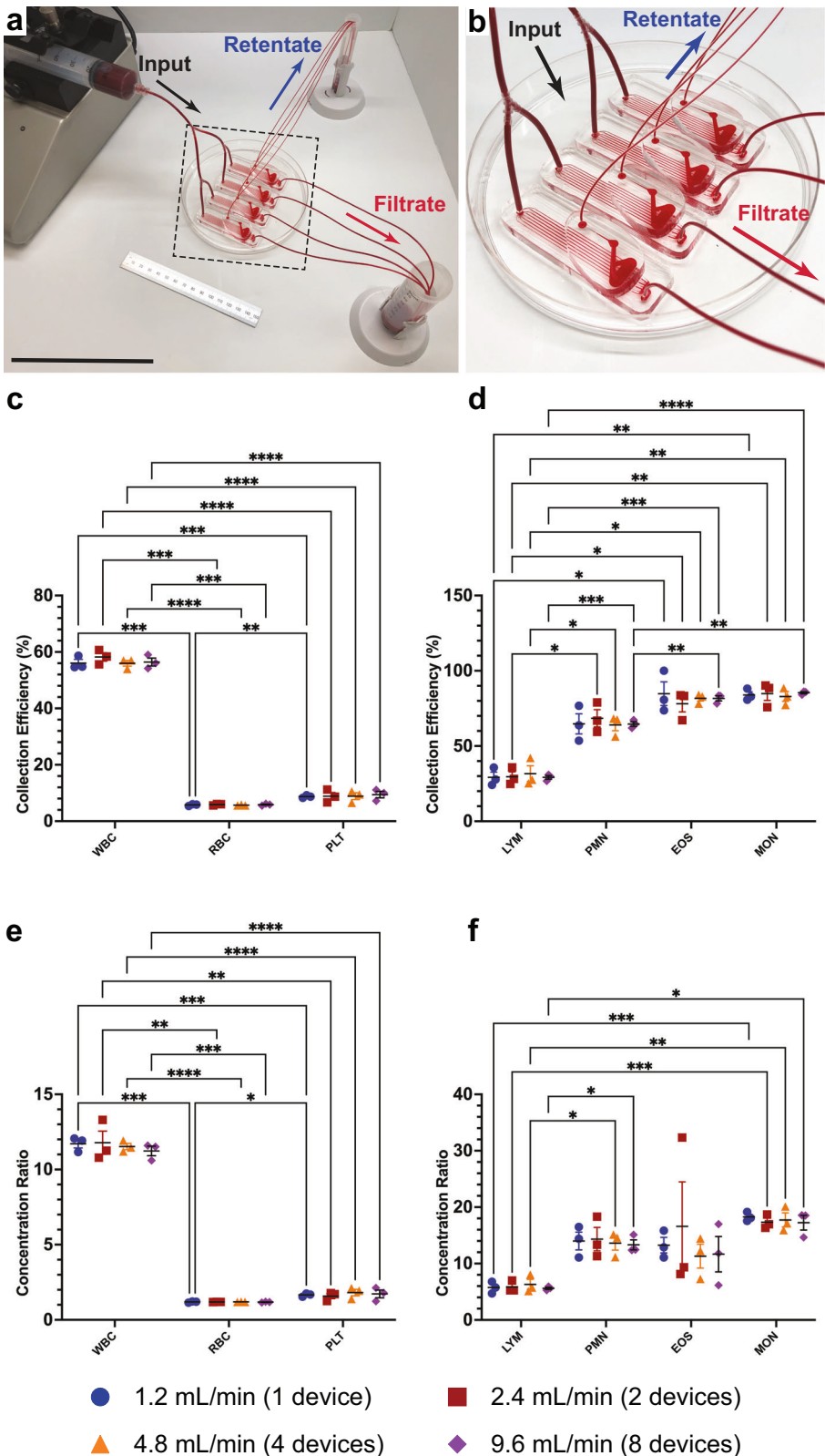

**Fig. 3 | Schematics, efficiency, and concentration ratios from parallelized in vitro experiments. a** Overall and (**b**) magnified images showing a representation of a 4-CIF device setup using Y-connectors to connect the inputs, with the outputs collected into either a retentate or a filtrate tube. Scale bar = 15 cm. **c**, **d** Increasing the number of devices in parallel did not affect the collection efficiency of (**c**) major cell types or (**d**) WBC subsets. **c** The CIF devices still had significantly higher WBC CE as compared to RBC and PLT, with (**d**) the larger WBC subsets being more efficiently collected. **e**, **f** The concentration ratios of the (**e**) major cell types or (**f**) WBC subsets were also not affected by increasing the number of devices in parallel. **e**, **f** Like efficiency, the collection ratio was the highest for the (**e**) WBC as compared to RBC and PLT and the (**f**) MON. Data shown as mean ± s.e.m. $n = 3$ repeated measures groups. Data analyzed by 2-way RM ANOVA with Tukey's multiple comparison test. *$p < 0.05$, **$p < 0.01$, ***$p < 0.001$, ****$p < 0.0001$.

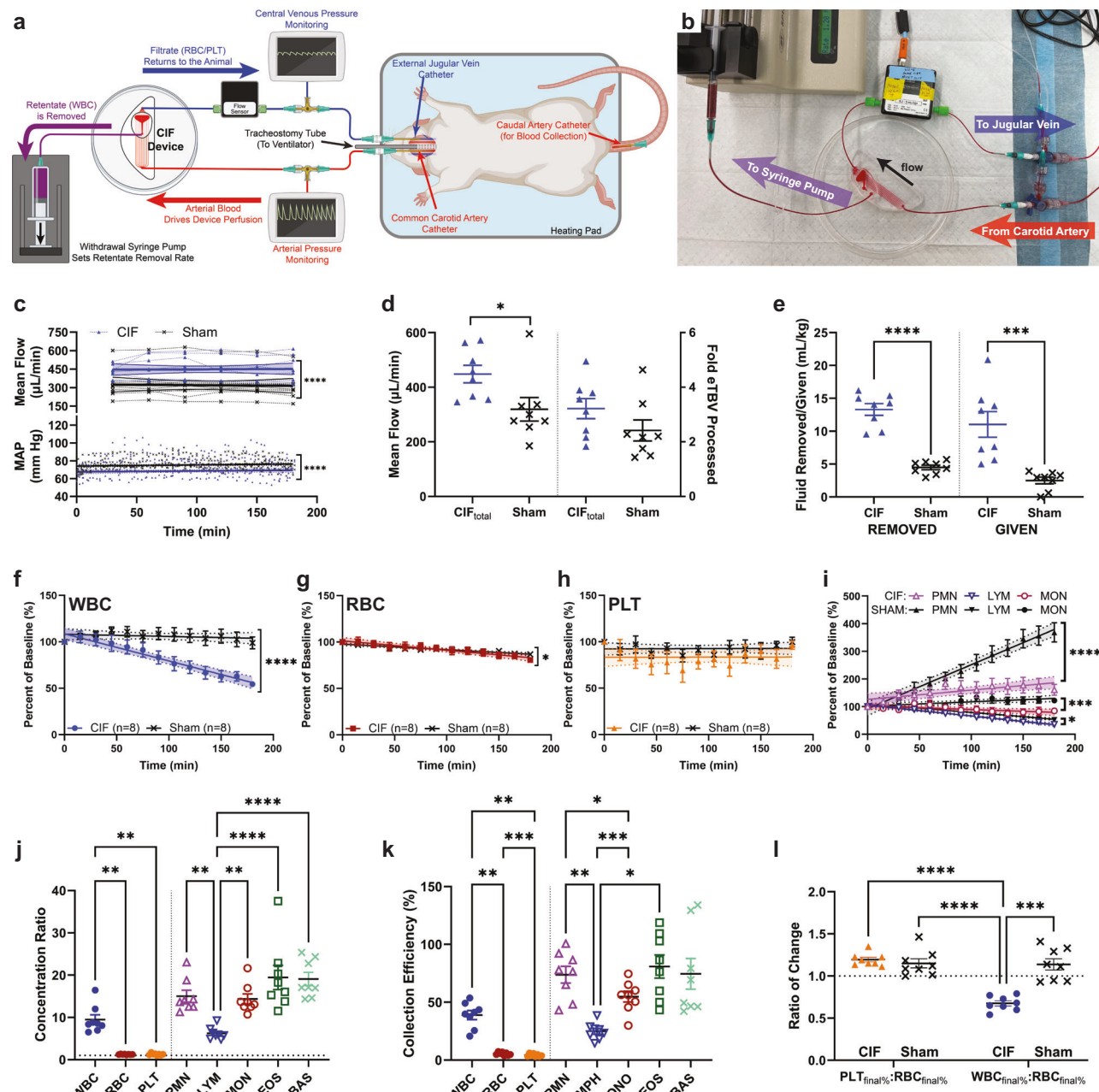

**Fig. 4 | Schematics, hemodynamics, and trends in blood counts of the in vivo recirculation experiments. a** Schematic (partially created in BioRender. Iqbal, M. (2024) https://BioRender.com/i31i968) and (**b**) photograph of the circuit in rats. **c** Although animals in the CIF group (▲) had an overall lower mean arterial blood pressure (MAP; $p < 0.0001$) and higher flow ($p < 0.0001$) throughout the procedure than those in the sham group (✗), there were no differences in the trend (slope) of MAP ($p = 0.757$) or flow ($p = 0.8076$) between the groups. **d** The overall mean flow rate was higher in the CIF as compared to the sham group ($p = 0.0303$); however, the estimated total blood volume processed was similar between both groups ($p = 0.1565$). **e** Rats in the CIF group had higher amounts of fluid removed ($p < 0.0001$) and given ($p = 0.0008$) than sham animals, although these were not associated with the percentage of WBC removed (Supplementary Table 2). **f**–**h** Rats in the CIF group had a significantly large decrease in circulating WBC (●, $p < 0.0001$) after 3 h, a small decrease in final RBC (■, $p = 0.0313$), and no difference in PLT (▲, $p = 0.9713$) as compared to sham-treated animals. **i** Throughout the 3 h experiment, rats in the CIF group had a lower rise in PMN (△, $p < 0.0001$), LYM (▽, $p = 0.0125$), and MON (○, $p = 0.004$) than those in the sham group (▲, ▼, ● for PMN, LYM, and MON, respectively). Overall (**j**) concentration ratio (CR) and (**k**) collection efficiency (CE) were significantly higher for WBC as compared to RBC and PLT, with both CR and CE being lowest for the LYM subset. **l** The ratio of change of PLT and RBC counts were similar between groups ($p = 0.9373$), whereas there was a significant difference in the ratio of change of WBC and RBC counts between the CIF and sham animals ($p = 0.0001$). Data shown as mean ± s.e.m. $n = 8$ per group. Data analyzed by (**c**) 2-tailed unpaired t-test for overall mean values and linear regression for slope, (**d**, **e**) 2-tailed unpaired t-test for overall mean values, (**i**) linear regression for slope, (**j**, **k**) 1-way RM ANOVA with Sidak's multiple comparison test, and (**l**) 2-way RM ANOVA with Sidak's multiple comparison test. *$p < 0.05$, **$p < 0.01$, ***$p < 0.001$, ****$p < 0.0001$.

both CIF and sham groups were unchanged (Fig. 5n). These data suggest that the CIF device itself does not induce a consumptive coagulopathy, with the rise in VWF and TAT most likely due to the surgical procedures themselves.

Furthermore, to determine if there was immune activation from the CIF device, we measured cytokine and chemokine profiles, comparing the CIF and sham groups before and after the leukapheresis procedures (Fig. 6). We found that the extracorporeal circulation

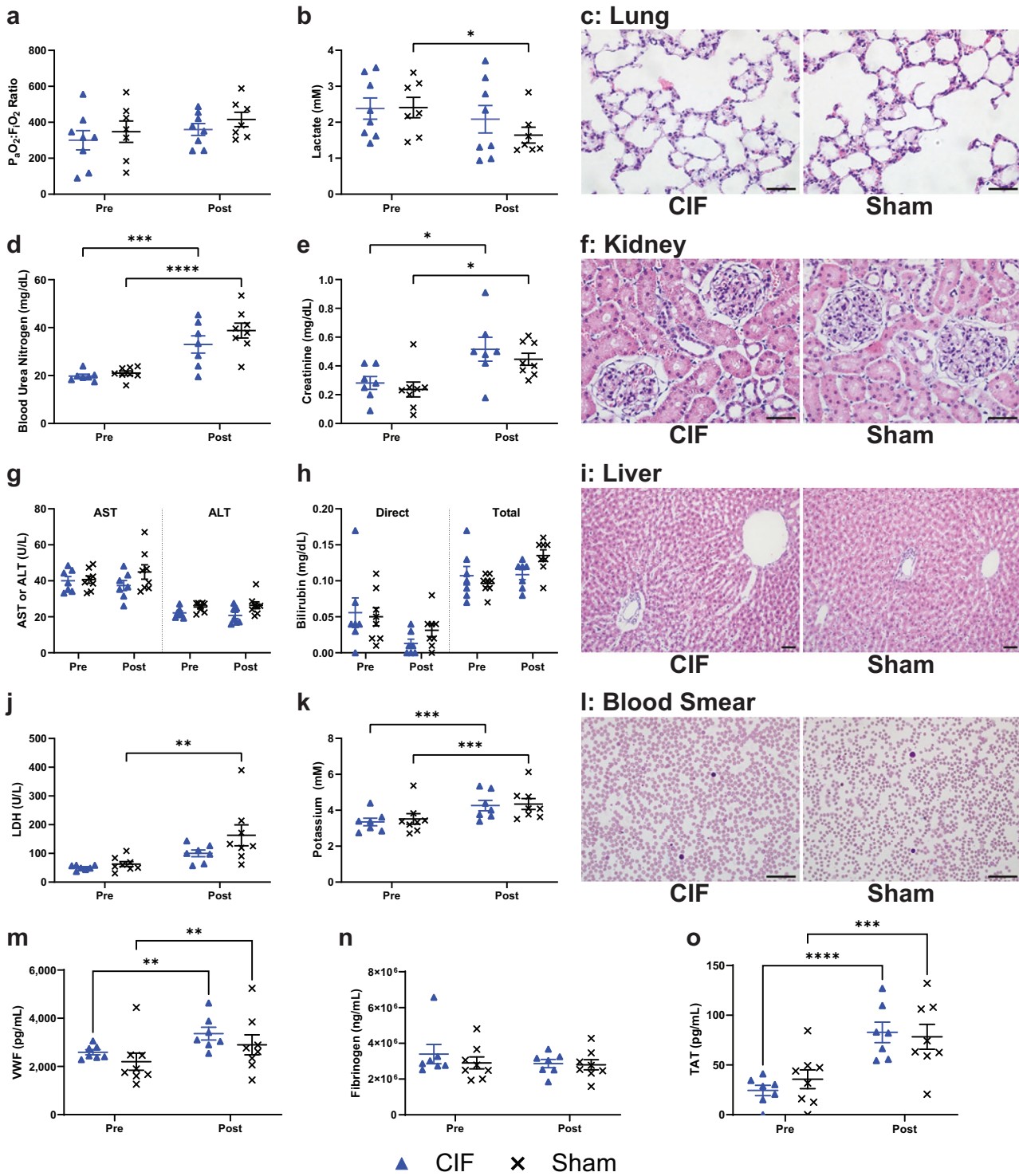

**Fig. 5 | End-organ safety in microfluidic leukapheresis.** Clinical laboratory data and representative histology comparing before (Pre) and after (Post) microfluidic leukapheresis procedures to assess for (**a–c**) cardiopulmonary, (**d–f**) renal, and (**g–i**) liver injury, markers of hemolysis (**j–l**), and (**m–o**) markers of coagulation system activation in CIF (▲) and sham (✕) groups. There were no differences between CIF- and sham-treated animals before and after the experiments. However, in both groups, there was a rise in blood urea nitrogen ($p = 0.001$ and $p < 0.0001$ for CIF and sham groups, respectively) and creatinine ($p = 0.0167$ and $p = 0.0214$ for CIF and sham groups, respectively) and a small, but clinically insignificant, rise in potassium ($p = 0.0002$ and $p = 0.0003$ for CIF and sham groups, respectively). In

the sham group only, there was a rise in lactate dehydrogenase (LDH; $p = 0.1539$ and $p = 0.0026$ for CIF and sham groups, respectively) and a small decrease in lactate ($p = 0.465$ and $p = 0.0305$ for CIF and sham groups, respectively). H&E sections appeared similar between the groups. In both groups, there was rise in von Willebrand factor (VWF; $p = 0.0015$ and $p = 0.0018$ for CIF and sham groups, respectively) and thrombin-antithrombin (TAT; $p < 0.0001$ and $p = 0.0001$ for CIF and sham groups, respectively) complexes, but no differences between the CIF and sham groups. Data shown as mean ± s.e.m. $n = 7$ for CIF and $n = 8$ for sham groups. Data analyzed using 2-way RM ANOVA with Sidak's multiple comparison test. $*p < 0.05$, $**p < 0.01$, $***p < 0.001$, $****p < 0.0001$. Scale bar = 50 μm.

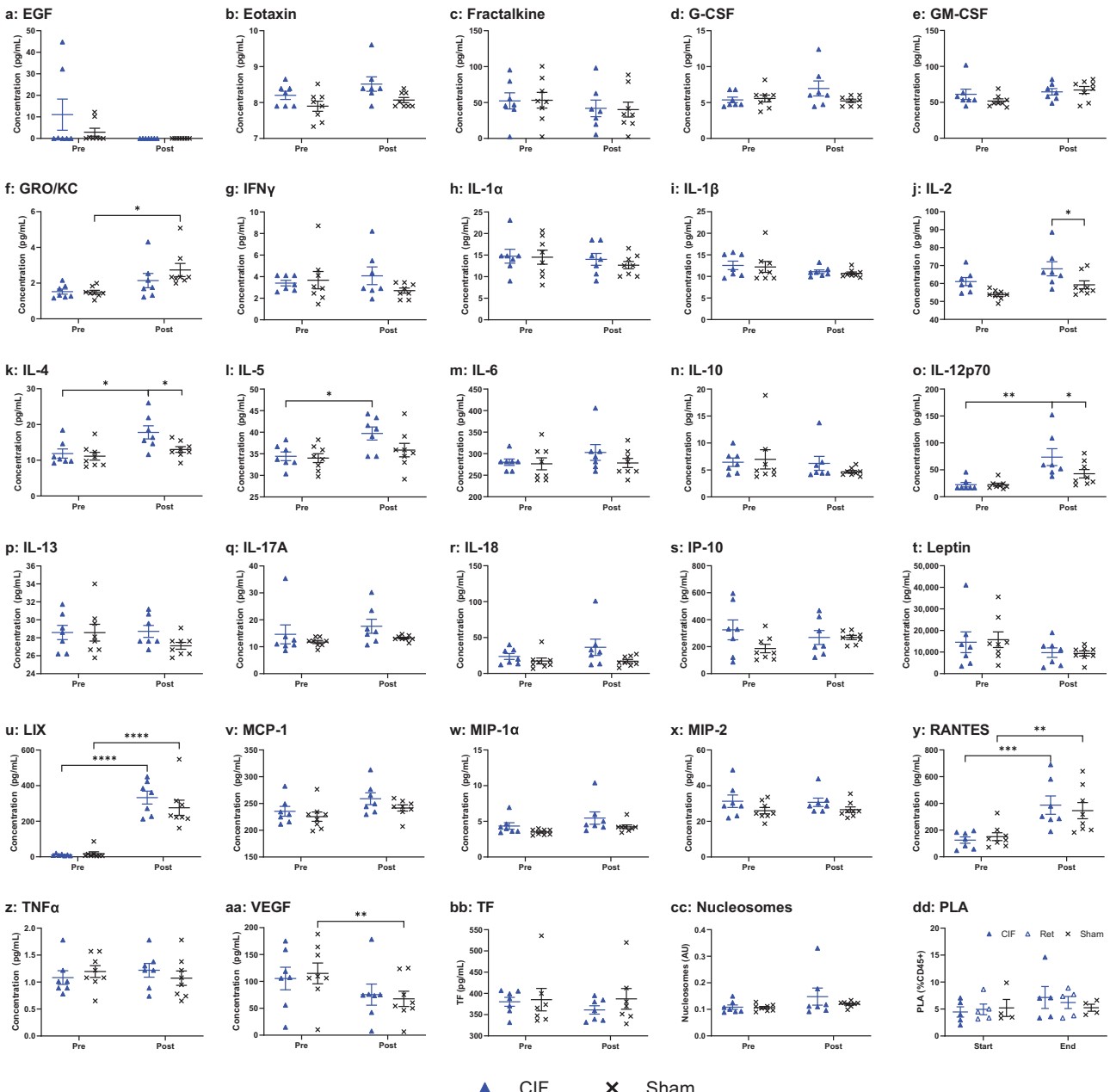

**Fig. 6 | Markers of inflammation in the blood before and after microfluidic leukapheresis procedures and sham-treated animals. a–aa** A 27-plex cytokine panel showed that, overall, plasma levels of most cytokines tested were similar between CIF (▲)- and sham (✕)-treated animals and before and after the procedures. Only (**j**) IL-2 ($p = 0.0335$), (**k**) IL-4 ($p = 0.0264$), and (**o**) IL-12p70 ($p = 0.0393$) were higher in CIF-treated animals as compared to sham controls after the procedures. In both groups, there was a rise in (**u**) LIX (CXCL5; $p < 0.0001$ and $p < 0.0001$ for CIF and sham groups, respectively) and (**y**) RANTES (CCL5; $p = 0.0007$ and $p = 0.0046$ for CIF and sham groups, respectively) after recirculation. In the CIF group alone, there was an increase in (**k**) IL-4 ($p = 0.0164$), (**l**) IL-5 ($p = 0.0138$), and

(**o**) IL–12p70 ($p = 0.005$), whereas in the sham group alone, there was a rise in (**f**) GRO/KC ($p = 0.0195$) and a decrease in (**aa**) VEGF ($p = 0.0035$) levels after recirculation. There were no differences in plasma (**bb**) tissue factor or (**cc**) nucleosome levels between or within the groups. **dd** There were no differences in the platelet-leukocyte aggregate (PLA) percentage between the groups or before and after the procedures. The retentate removed by the CIF device had similar levels of PLA as the recirculating blood in both groups. Data shown as mean ± s.e.m. $n = 7$ for CIF and $n = 8$ for sham groups. Data analyzed using 2-way RM ANOVA with Sidak's multiple comparison test. *$p < 0.05$, **$p < 0.01$, ***$p < 0.001$, ****$p < 0.0001$.

induced a small degree of inflammation in both groups, although the changes were inconsistent. Specifically, there was a rise in CXCL5/LIX (Fig. 6u) and CCL5/RANTES (Fig. 6y) in both groups, whereas IL-4 (Fig. 6k), IL-5 (Fig. 6l), and IL-12p70 (Fig. 6o) rose only in the CIF group. In addition to changes in CXCL5 and CCL5, animals in the sham group had higher CXCL1 (GRO/KC) and lower VEGF after the procedures

(Fig. 6f,aa). Additionally, we measured plasma nucleosomes as a surrogate marker for PMN activation[42,43] and found no differences between the groups (Fig. 6cc). Finally, imaging flow cytometry found no difference in PLT-WBC aggregate formation[44–46] between the CIF or the sham groups nor the CIF device retentate containing WBC removed from the animal during the procedures (Fig. 6dd).

## Discussion

In this report, we describe a microfluidic device capable of separating WBC from undiluted WB with throughput and efficiency sufficiently high to enable leukapheresis in a live animal. Our data show that the CIF devices were safe, well-tolerated in animals, and could operate in the recirculation regime in vivo for at least 3 h after a single dose of heparin anticoagulant without any loss of separation function. Designed to separate the largest WBC, CIF devices showed the highest collection efficiency for eosinophils (76–87%) and monocytes (73–85%) for normal human blood in vitro, and for eosinophils (82%) and neutrophils (74%) for rat blood in vivo. Our parallelization experiments demonstrated CIF's ability to operate at high throughput of ~10 mL min$^{-1}$ and deliver consistently high performance, as shown by the human monocyte and eosinophil collection efficiency of ~85%. Furthermore, when tested in vitro using blood samples spiked with a human leukemia cell line, the CIF devices showed a high CE for the leukemic blasts (73–90%, depending on device design) that compared favorably to the values (47–54%, depending on the study)[47,48] reported recently for centrifugation-based therapeutic leukapheresis in leukemia patients. Nevertheless, further testing is required to confirm that this remarkably high CE demonstrated by the CIF devices will hold true for blood samples of children with acute leukemia.

Our data in rats show the potential benefits of CIF-enabled microfluidic leukapheresis over the currently used centrifugation-based method. First, our rat model shows that microfluidic leukapheresis can occur without the need for a driving pump to achieve a similar amount of blood processing (~1 TBV per hour) and leukocyte removal (~46%) as performed in children at our institution. This could greatly simplify leukapheresis by reducing the need for the large bore venous catheters typically used to perform centrifugation-based leukapheresis. Leukapheresis could potentially be performed using peripheral artery to vein connections that typically require less sedation to place. Second, this amount of blood processing is possible using a system with minimal ECV. Third, microfluidic leukapheresis did not incur organ injury in our rat model. The slight rise in creatinine and blood urea nitrogen occurred in both the CIF and sham groups and likely represented repeated blood collection for serial CBC measurements. Finally, microfluidic leukapheresis had a minimal effect on PLT activation or inflammation. Taken together, these data suggest that CIF-enabled microfluidic leukapheresis may be safe to use in humans in lieu of other, more invasive, methods. However, longitudinal follow-up studies (weeks/months post-treatment) in large animal models are necessary to ensure a lack of negative long-term sequelae from CIF-based leukapheresis.

Although intended to remove only the largest WBC, the CIF devices were able to reduce the overall WBC count in live animals by 46% after processing ~3.3 eTBV. This reduction was on par with our observations in children (50% WBC reduction after processing ~3.6 eTBV) and the 31–50% (1.2–2.3 eTBV processed) WBC reductions reported in the literature for centrifugation-based leukapheresis[47,48]. However, there is a drastic difference in total eTBV between the rats and humans, and, therefore, future studies will focus on testing the CIF device performance in a large animal model (e.g., pig) with eTBV closely matching that of pediatric patients, before ultimately proceeding to the first-in-human trials.

We noted a gradual decrease in RBC count over time in both CIF and sham control groups, most likely due to frequent blood sampling during the procedure. However, the decrease in WBC count in the CIF group was greater than the decrease in RBC count, suggesting that we were able to selectively remove WBC from circulation using the device. Importantly, there was no significant loss of PLT after the microfluidic leukapheresis in vivo, in contrast to the 32% reduction in PLT count we observed for centrifugation-based leukapheresis in our pediatric cohort and the 23–31% PLT loss reported in the literature[48,49]. Such a dramatic difference is likely because CIF separates cells based on size, and there is significantly less overlap between WBC and PLT in terms of size than density[50,51]. Preventing PLT loss during the leukapheresis procedure could improve its safety and reduce the need for PLT transfusions.

The ECV of the microfluidic leukapheresis circuit in our animal experiments was 0.74 mL, of which only a small fraction was due to the dead volume of the CIF device (<0.1 mL). In our parallelized eight-device version (64 CIF elements, Supplementary Fig. 4c) the device and the tubing had a dead volume of ~4.2 mL, which is about 1/70th of the ~300 mL ECV of a typical centrifugation-based leukapheresis circuit currently used in clinical practice. Such a dramatic reduction of ECV would be particularly important in leukemic infants with hyperleukocytosis who are often too small on which to perform centrifugation-based leukapheresis safely[14], such as in our clinical cohort where the smallest patient receiving leukapheresis was 9 kg (7.2 months old). Because of their small size, they required mechanical ventilation for placement of the large bore catheter, circuit blood priming, and became hypotensive during leukapheresis. A smaller device with similar efficacy, like CIF-based leukapheresis, could potentially mitigate most of these adverse effects or need for additional resources. For example, our multiplexed eight-device system ran at 9.6 mL min$^{-1}$ (576 mL hr$^{-1}$). For a 9 kg patient, that flow rate would result in a processing rate of 0.91 eTBV per hour which is on par with our current clinical practice (0.8 [0.6–1.0] eTBV per hour, Fig. 1). Importantly, ECV of our eight-device multiplex would represent only 0.7% of the patient's eTBV, which is drastically lower than for centrifugation-based leukapheresis (~48% of the patient's eTBV).

During our animal experiments, the flow of blood through the CIF device was driven by the arteriovenous pressure difference alone. Pumpless systems are common in other extracorporeal therapies, such as lung assist[52,53]. A major benefit of pumpless systems is that they reduce the risk of activation of PLT and WBC[30] and reduce the amount of anticoagulation needed (e.g., our experiments required only a single dose of heparin as opposed to the continuous citrate infusions typically used in centrifugation-based leukapheresis). Although our in vivo system was not completely "pumpless" due to the use of a syringe pump to precisely control the WBC removal rate, blood returning to the animal was not directly exposed to a pump. If precise, dynamic control of the removal rate is not required, then operation of the CIF device could be made entirely passive by including properly balanced fluidic resistors downstream[32]. Of course, the CIF devices could also be operated with a perfusion pump as well[30,32], although with an increasing risk of PLT and WBC activation as well as cytolysis, which will depend on the characteristics of the pump used.

Although leukapheresis has been described as safe and effective in acute leukemias[8,54–57], it remains a controversial practice and its use in children in the United States has been reportedly decreasing[58]. The main concerns include the risks associated with the preparation for (and process of) leukapheresis[11,12,59,60] and the potential delay in the initiation of definitive treatment with chemotherapy, with some centers preferring low-dose chemotherapy for cytoreduction over centrifugation leukapheresis[3,33]. Our clinical experience in pediatric patients showed that centrifugation leukapheresis reduced PLT counts, required many patients to be deeply sedated, and necessitated RBC priming of the circuit due to complications arising from the ECV being a significant percentage of their eTBV. Taken together, our data suggest that CIF-based microfluidic devices may be a feasible alternative to centrifugation-based leukapheresis in pediatric patients with acute leukemias. Ultra-low ECV, high collection efficiency for leukemic blasts, minimal PLT loss, and the ability to operate effectively at lower flow rates offered by CIF devices may obviate most of the drawbacks associated with current centrifugation-based procedure. This simplified process may be especially useful in children to reduce the complications in the preparation for and process of leukapheresis as well as to shorten the time to leukapheresis and, ultimately, chemotherapy.

## Methods

### Ethics statement

Human subjects research was approved through the Institutional Review Board at Baylor College of Medicine (H-3342) and the University of Houston (16272-01). Animal subjects research was approved through the Institutional Animal Care and Use Committees at Baylor College of Medicine (AN-8224) and Michael E. DeBakey Veterans Affairs Medical Center (20K05.A).

### Clinical leukapheresis data collection

To understand the current workflow and WBC removal efficiency of centrifugal leukapheresis for hyperleukocytosis in leukemic children, we assessed 43 pediatric leukapheresis procedures performed in 37 patients from 2/2011 to 12/2022 in a large, urban, quaternary pediatric hospital. Leukapheresis was performed using either the COBE® Spectra (28 procedures; 2011–2016) or Spectra Optia® (15 procedures; 2018–2022) systems (Terumo). The extracorporeal volumes (ECV) of the circuits, including a blood warmer, were ~300 [289–321] mL. Leukapheresis procedures were excluded if data collection regarding the leukapheresis procedure itself was incomplete. Data collected included the time between leukapheresis and hospital admission, blood volume processed, complications, complete blood count (CBC) measurements, blood transfusions within 2 h of leukapheresis, blood pressure, and the need for critical care services (e.g., intubation and mechanical ventilation solely for the purpose of placing vascular catheters and/or performing the leukapheresis procedure, initiation or uptitration of cardiovascular support medications, and need for monitored anesthesia care). Baseline CBC values were those collected within the prior 3 h of leukapheresis start or the closest one after initiation (the median time was −20 min [−57 to −10] from leukapheresis start). Estimated total blood volume (eTBV) was calculated assuming 70 mL kg$^{-1}$, and the processed eTBV was calculated by the formula:

$$\frac{\text{inlet flow volume (mL)} - \text{anticoagulation flow volume (mL)}}{\text{eTBV}} \quad (1)$$

### CIF device development

We described the operating principle of 'controlled incremental filtration' (CIF) and the mathematical framework used for creating a CIF design in detail previously[27,28]. Briefly, a typical CIF element consists of one middle (retentate) channel and two symmetrical side (filtrate) channels divided by a series of slanted, pill-shaped posts (Fig. 2a, Supplementary Fig. 1), separated from one another by fixed-size gaps (~20 μm). At each filtration gap $i$, a fraction ($f_{gap}(i)$) of the retentate channel flow ($Q_r(i)$) is directed into either filtrate channel ($Q_{gap}(i) = f_{gap}(i) * Q_r(i)$), and as a result, the flow in the retentate channel decreases (and in filtrate channels increases) with increasing $i$. The width of the fluid lamina ($w_l$) passing through each gap determines the size of the cells that are small enough to be carried by the extracted fluid into the filtrate channels (Fig. 2a)[32]. When designing a CIF element, the initial values of the filtration fraction ($f_{gap}*$) and retentate channel width ($w_r*$) are selected based on previous empirical data[28,32]. The filtration fraction ($f_{gap}(i)$) and the width of the retentate ($w_r(i)$) and/or filtrate ($w_f(i)$) channels are then iteratively recalculated (using the previously-described mathematical framework[27,28]) to maintain the extracted fluid lamina width ($w_l$) and therefore the size cutoff constant along the length of the CIF element.

The theoretical maximum number of cells smaller than the size cutoff that could be recovered in the filtrate is determined by the flow ratio (FR; the collected volume ratio of the filtrate and retentate) of the CIF element. For example, a CIF element with a FR = 20 would have approximately 1/(1 + FR) = 4.8% of the cells below the size cutoff lost to the retentate output. Similarly, FR limited the maximum degree to which cells larger than the size cutoff could be concentrated in the retentate. For example, a CIF element with a FR = 20 could increase the retained cell concentration by up to 21-fold (1 + FR).

For this study, we designed four CIF elements with same overall architecture and $w_r*$, but with different $f_{gap}*$: $0.72 \times 10^{-3}$ (Design 1), $1.00 \times 10^{-3}$ (Design 2), $1.28 \times 10^{-3}$ (Design 3), and $1.70 \times 10^{-3}$ (Design 4). Each CIF device consisted of a 'device layer' containing eight identical CIF elements of the same design arranged in parallel. The device layer had one input that bifurcated into eight elements, and one filtrate output from the converging filtrate channel outputs of the eight elements (Supplementary Fig. 1). Since the retentate channels are surrounded by filtrate channels (Supplementary Fig. 1a, b), they cannot be connected within the device layer. Therefore, a separate manifold/top layer collected the outputs from the eight retentate channels in the device layer and directed it to a single output (Fig. 2a–c, Supplementary Movie 2).

The device designs were generated using MATLAB (The MathWorks Inc; Natick, MA) and transferred to a chrome-on-glass photomask (Photo Sciences; Torrance, CA). Using a ~150 μm-thick SUEX photoresist (DJ MicroLaminates Inc.; Sudbury, MA) film, the designs were reproduced on a 4" silicon wafer (UniversityWafer Inc.; Boston, MA) through photolithography, resulting in one wafer for the device layer and one for the top layer. These wafers acted as master molds for producing the devices from poly(dimethylsiloxane) (PDMS) (Sylgard 184; DOW, Midland, MI). The PDMS replicates were cured at 69 °C for 20 h, after which holes were made using biopsy punches (Robbins Instruments, Houston, TX) for the manifold and tubing connections. The device layer was bonded to a PDMS flat layer acting as a base, and the manifold layer was bonded to the device layer, using a plasma oxidizer (PDC-001; Harrick Plasma; Ithaca, NY; Fig. 2b,c). The assembled devices were treated with 1% (w/v) aqueous solution of mPEG-silane (MW 5000 Da; Laysan Bio Inc; Arab, AL) for 20 min at 69 °C, followed by a flush and incubation with 1% BSA buffer solution overnight to minimize cellular adhesion and consequent device clogging (Supplementary Fig. 7). When performing animal experiments, devices were additionally flushed and incubated (for at least 30 min) with heparinized saline (10 U mL$^{-1}$) before each run. The assembled and treated devices were stored at 2–8 °C until use. Each device had a small footprint (39 cm²) and minimal dead volume (~87 μL). For in vitro experiments, the input, retentate, and filtrate output tubing added a further dead volume of ~214 μL, for a total of ~301 μL.

### Human blood collection

Whole blood (WB) units, anticoagulated in citrate phosphate dextrose (CPD), were purchased from Gulf Coast Regional Blood Center (Houston, TX) and filtered using a 40-μm aggregate filter (SQ40; Haemonetics, Boston, MA) before being run through the devices. Additionally, fresh WB was obtained from healthy adult volunteers via venipuncture and collected in ACD-A tubes (BD, Franklin Lakes, NJ).

### Leukemic blood cell culture and sample preparation

The human AML cell line MV-4-11 (ATCC)[61] was cultured in RPMI with L-glutamine (HyClone; Marlsborough, MA), 10% bovine growth serum (HyClone; Marlsborough, MA), and 1% penicillin/streptomycin (Gibco; Waltham, MA). The culture was maintained in a humidified 37 °C/5% CO$_2$ incubator (In-VitroCell ES NU-5810, NuAire, Plymouth, MN) and cells were passaged every 48 h[62,63].

To test CIF devices' efficiency in separating human leukemia cells, we spiked normal human WB with the MV-4-11 cells. Cultured cancer cells were centrifuged at 300 × g, for 5 min, washed in phosphate-buffered saline (PBS; HyClone; Marlborough, MA), and resuspended in PBS for a concentration of 1 × 10⁶ per mL. The cells were then stained with Vybrant DiI (Invitrogen; Waltham, MA) dye, incubated at 37 °C for 20 min, washed twice with PBS, and then added to the whole blood.

## Collection efficiency calculation

Generally, collection efficiency for a CIF device is defined as follows:

$$\frac{\text{Cells collected in retentate}}{\text{Total cells processed through the device}} \times 100 \quad (2)$$

Whereas calculating cells collected in retentate is straightforward, the formula for calculating total cells processed through the device can differ based on the experimental setup (flow-through vs recirculation) and data collected. These formulas have been defined in the appropriate sections below.

## In-vitro experiments

For normal human WB experiments, samples were loaded into a 10 mL syringe (BD, Franklin Lakes, NJ) and infused into the device using a syringe pump (Genie Touch, Kent Scientific, Torrington, CT) through medical grade polyethylene tubing (BB31695-PE/6; ID 0.86 mm, OD 1.52 mm; Scientific Commodities Inc.). Using polyurethane tubing, the retentate (BB520-25; ID 0.305 mm, OD 0.635 mm; Scientific Commodities Inc.) and the filtrate (BB520-40; ID 0.635 mm, OD 1.02 mm; Scientific Commodities Inc.) samples were collected in polypropylene vials (CELLTREAT Scientific Products; Pepperell, MA). CBC for the input, retentate, and filtrate blood samples were measured using the Sysmex XS-1000i (Sysmex, Japan) hematology analyzer immediately after each experiment. The ability to concentrate cells in the retentate channel (CR, concentration ratio) was calculated as the ratio of cell concentrations in the retentate channel over the input sample. To account for the retentate and filtrate output volumes, device performance was also evaluated by calculating the collection efficiency (CE) of each CIF design using the following formula:

$$\frac{C_{\text{ret}}}{\left(C_{\text{fil}} \times \frac{V_{\text{fil}}}{V_{\text{ret}}}\right) + C_{\text{ret}}} \times 100 \quad (3)$$

Where $C_{\text{ret}}$ = cell concentration in the retentate (cells per μL)

$C_{\text{fil}}$ = cell concentration in the filtrate (cells per μL)

$V_{\text{fil}}$ = volume collected from the filtrate output (μL)

$V_{\text{ret}}$ = volume collected from the retentate output (μL).

Human WB samples spiked with fluorescently labeled MV-4-11 cells were also processed similarly. Spiked blood samples were loaded into a syringe and infused into the device using a syringe pump. The concentrations of MV-4-11 cells in the input, retentate output, and filtrate output samples were obtained by manually counting the cells using a hemocytometer (In-cyto, South Korea) with a Neuber-Improved grid and a fluorescent microscope (IX71, Olympus, Japan) (see Supplementary Fig. 3). The collection efficiency for the cancer cells was calculated using the following formula:

$$\frac{MV411_{\text{ret}} \times V_{\text{ret}}}{MV411_{\text{input}} \times (V_{\text{ret}} + V_{\text{fil}})} \times 100 \quad (4)$$

Where $MV411_{\text{ret}}$ = MV-4-11 concentration in the retentate (cells per μL)

$MV411_{\text{input}}$ = MV-4-11 concentration in the input (cells per μL).

## In-vitro multiplexing experiments

We connected two, four, and eight Design 2 devices in parallel using 1/16″ Y-connectors (Fisherbrand, Waltham, MA) and polyurethane tubing (BB520-95, Scientific Commodities Inc., Lake Havasu, AZ) (see Fig. 3 and Supplementary Fig. 4). For comparison, we also ran one Design 2 device. Undiluted WB was loaded into 10, 30, or 50 mL syringes and ran through the devices using a syringe pump (Kent Scientific). All retentate tubing (BB520-25, Scientific Commodities Inc.) was directed to a single polypropylene collection vial (CELLTREAT); similarly, all filtrate tubing (BB520-40, Scientific Commodities Inc.) was directed to a separate single collection vial (CELLTREAT).

## Animal anesthesia and ventilation

Male and female Sprague-Dawley outbred rats (Crl:CD(SD); Charles River, Wilmington, MA), ages 10–16 weeks, were used in this study. Rats were induced under a surgical plane of anesthesia using inhaled isoflurane and positioned supine on an infrared heating pad with a target rectal temperature of 37 °C. After sterile preparation of the neck, rats were given polyethylene tracheostomy tubes (PE240; ID 1.68 mm, OD 2.42 mm; Intramedic; BD; Franklin Lakes, NJ) for mechanical ventilation and maintenance of anesthesia with 100% oxygen and inhaled isoflurane (1.0–3.5%; Somnosuite; Kent Scientific). Animals were ventilated with ~6.75 mL kg$^{-1}$ tidal volume, zero end expiratory pressure, and a respiratory rate of ~70 breaths per minute. Ventilation was adjusted to maintain a goal paCO$_2$ ~35–40 mm Hg (correlated to end-tidal CO$_2$ measurements; etCO$_2$) and a goal SaO$_2$ > 95% (correlated to pulse oximetry; SpO$_2$). Ventilator and blood pressure data were recorded using a Hewlett Packard 78353B monitor (Spring, TX) or PowerLab data acquisition system and analyzed in LabChart 8 Pro (ADInstruments; Colorado Springs, CO).

## Vascular cannulation

The left external jugular vein (LEJV), right common carotid artery (RCCA), and ventral caudal artery (VCA) were cannulated like in previously published protocols[64–66]. Once the LEJV and RCCA were isolated, the cranial ends of both vessels were ligated with sutures and 22-gauge peripheral intravenous catheters (1″ length; Smiths Medical ASD, Inc.; Minneapolis, MN) were inserted into the vessels and secured in place. Once the neck was sutured closed, the tail was sterilely prepared for ventral caudal artery cannulation to serially collect blood without disrupting the blood flow through the extracorporeal circuit. Once isolated, the caudal end of the VCA was ligated and a catheter made of polyethylene tubing was inserted (PE10; ID 0.28 mm, OD 0.61 mm; Intramedic; BD; Franklin Lakes, NJ). All catheters were flushed with 10 U mL$^{-1}$ heparinized saline (diluted in 0.9% sodium chloride) to prevent thrombosis. After insertion of the cannulas, rats were given an intravenous bolus of 500 U kg$^{-1}$ heparin sodium (100 U mL$^{-1}$; Medefil Inc, Glendale Heights, IL), which was allowed to circulate for 5 min prior to initiation of extracorporeal circulation to minimize thrombosis[37–39].

## Extracorporeal circulation and leukapheresis

The extracorporeal circuit for leukapheresis was set up as shown in Fig. 4a, b. A CIF device (~87 μL; CIF group) or a resistance bridge (sham group) was connected between the RCCA and LEJV using polyurethane tubing (PU-065; ID 0.76 mm, OD 1.65 mm; SAI Infusion Technologies; Lake Villa, IL). Three-way stopcocks (260 μL; B. Braun Medical) open to all sides were used to measure the arterial and venous blood pressures during experiments. Arterial blood pressure was the driving force perfusing the CIF device or bridge. An inline flowmeter (80 μL; Elveflow MFS 5; Elvesys; Paris, France) was placed after the CIF device or bridge to measure the blood flow returning to the animal (filtrate). The estimated total ECV with the CIF device, tubing, and connections was ~736 μL, accounting for ~3% of the rat's eTBV. In the CIF group, separated WBC were collected into 1-mL heparinized syringes using a withdrawal syringe pump connected to the retentate channel via PU-065 tubing, with the flow rate set to ~5% of the measured flow rate of the filtrate output returning to the animal (FR of ~20). The mean flow rate was expressed as 30 min epochs (e.g., 0–30 min, 30–60 min, 60–90 min, etc.) by adding the mean filtrate flow rate to the calculated weighted mean retentate flow rate (volume collected divided by time of collection) for each 30 min epoch.

Once the extracorporeal circuit was attached, flow was opened to the CIF device or resistance bridge and the animal was allowed to equilibrate for 5 min to allow for any hemodynamic and/or dilutional changes to occur due to the ECV. Then, the baseline CBC was collected and, for animals in the CIF group, collection via the retentate channel

was initiated. Sodium chloride (0.9%; 0.5–1 mL aliquots) was given if the mean arterial pressure was <10% of the baseline value. Heparinized saline (10 U mL⁻¹; 0.1 mL) was given to flush tubing after blood collection or to keep the pressure lines clear. All extracorporeal circulation experiments were performed for 3 h prior to euthanasia.

## Measurement of laboratory values

Baseline arterial blood gas (ABG) and plasma were collected prior to flow initiation. Serial CBC were collected 15 min after the baseline CBC and every 15 min thereafter for 3 h. For rats in the CIF group, the retentate samples were collected every 30 min after retentate collection initiation to minimize disruption in WBC removal. The concentration ratio (CR) for each cell type was calculated for each timepoint using the following formula, with the overall concentration ratio calculated by the mean:

$$\frac{C_{ret_t}}{C_{fil}} \tag{5}$$

The collection efficiency (CE)[48] was calculated as follows:

$$\frac{\sum \left( C_{ret_t} \times V_{ret_t} \right)}{\left( \frac{C_{pre} + C_{post}}{2} \right) \times V_{proc}} \tag{6}$$

Where $C_{ret_t}$ = retentate concentration at timepoint $t$ (cells per μL)
$V_{ret}$ = retentate volume at timepoint t (μL)
$C_{pre}$ = peripheral blood concentration at experiment start (cells per μL)
$C_{post}$ = peripheral blood concentration at experiment end (cells per μL)
$V_{proc}$ = total blood volume processed (μL)
$C_{fil}$ = mean filtrate cell concentration during the time period (cell per μL).

Arterial blood gases with metabolic panels were measured using Element POC test cards (ePOC; Heska; Loveland, CO). $PaCO_2$ and $SaO_2$ were used to validate the end-tidal $CO_2$ (etCO₂) and pulse oximetry ($SpO_2$), respectively, as well as to measure WB lactate. CBC, including 5-part differential, were performed on an Element HT5 veterinary hematology analyzer (Heska). To obtain clinically relevant markers of organ injury, plasma was centrifuged at $1900 \times g$ for 15 min, aliquoted, and stored at −80 °C prior to measurement using a Beckman Coulter AU480 chemistry analyzer. Plasma samples were sent for cytokine and chemokine measurements which were performed in duplicate using the Rat Cytokine/Chemokine 27-Plex Discovery Array (Eve Technologies; Calgary, AB).

## Flow cytometry

We performed imaging flow cytometry (Amnis Imagestream^X Mk II, Luminex Corporation, Austin, TX) on a subset of experiments to assess the degree of PLT-WBC aggregates in CIF and sham-treated animals as a marker of cell activation[44,45,67]. WB (100 μL) was labeled with allophycocyanin-conjugated mouse anti-rat CD45 (0.5 μg per test; clone OX-1; Invitrogen), phycoerythrin-conjugated Armenian hamster anti-mouse/rat CD42d (1 μg per test; clone 1C2; Invitrogen), and fluorescein isothiocyanate-conjugated Armenian hamster anti-mouse/rat CD61 (1 μg per test; clone2C9.G3; Invitrogen) antibodies for 20 min at room temperature. Non-immune IgG control antibodies were used to assess for non-specific binding. Samples were then fixed with 2% paraformaldehyde and placed at 4 °C until analysis within 24 h. Data were analyzed in IDEAS 6.2 (Luminex Corporation) software. Compensated flow cytometry files were run through the IDEAS 6.2 co-localization wizard. PLT-WBC aggregates were defined as the percentage of WBC that had PLT within 6 μm of each other's center (see example shown in Supplementary Methods).

## Histology

In a subset of experiments, we created blood smears using the PRO-TOCOL Hema 3 manual staining pack (Fisher Healthcare). The smear slides were overlaid with a glass slide of similar dimensions containing a custom grid of twelve 0.5 mm squares. One image was taken per square per sample. Slides were imaged on an inverted brightfield microscope (IX83, Olympus, Japan) and the size measured using FIJI/ImageJ. In a subset of animals, hematoxylin and eosin (H & E) staining of the lungs, kidneys, and livers were performed to assess for gross differences between CIF and sham groups. After euthanasia, the animals were fix-perfused with 10% neutral buffered formalin using the LEJV and RCCA cannulas. Perfusion was performed until the effluent was clear. After opening of the thoracic and abdominal cavity, the lungs were inflated with 10% formalin via the tracheostomy tube at a pressure of ~25 cm $H_2O$[46,68]. After inflation, the trachea was tied to maintain inflation, and then the lungs were placed into 10% formalin. The kidneys and liver were sectioned and placed into 10% formalin. After 72 h of fixation, the tissues were processed and paraffin-embedded. H&E slides were prepared using 5 μm sections and imaged using an Olympus IX83 microscope with either a 20X/0.7 dry or 40X/1.25 silicone oil objective lens and captured using cellSense (Olympus).

## Data analysis

All statistical analyses were performed in Prism 10 (GraphPad Software, LLC). Parametric data are reported as mean ± standard error of the mean (s.e.m.) whereas non-parametric data are reported as median [interquartile range (IQR)]. Analyses between two independent groups were performed using Student's t-test or Mann-Whitney test for non-parametric data. Analyses comparing final or nadir blood count percentages to a baseline of 100% were performed using one sample Wilcoxon test with Pratt correction. Analyses between three or more independent groups were performed using 1- or 2-way analysis of variance (ANOVA) with Tukey's multiple comparisons test or Kruskal-Wallis tests with Dunn's multiple comparisons test. Repeated measures were analyzed using either paired t-test, 1-way repeated measures (RM) ANOVA, or 2-way RM ANOVA with Sidak's or Tukey's multiple comparisons test, where appropriate. Serial data over time were analyzed using simple linear regression analysis and reported as mean with 95% CI. Multiple linear regression analysis using the least square method was performed to assess which variables were independently associated with a change in cell concentrations. *$p < 0.05$, **$p < 0.01$, ***$p < 0.001$, & ****$p < 0.0001$.

## Reporting summary

Further information on research design is available in the Nature Portfolio Reporting Summary linked to this article.

# Data availability

All data supporting the findings of this study are available within the article and its supplementary files. Any additional requests for information can be directed to, and will be fulfilled by, the corresponding authors. Source data are provided with this paper.

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

## Acknowledgements

We would like to thank Dr. Alexandra Stevens (BCM) for providing the MV-4–11 cell line. Research reported in this publication was supported in part by the National Heart, Lung, and Blood Institute of the National Institutes of Health under Award Number R01HL151858. The content is solely the responsibility of the authors and does not necessarily represent the official views of the National Institutes of Health, the U.S. Department of Veterans Affairs, or the United States Government. A portion of Fig. 4a (rat image) and Supplementary Fig. 4 were created in BioRender. Iqbal, M. (2024) https://BioRender.com/i31i968 and edited.

## Author contributions

A.Mo., P.R.S., T.P.F., and F.W.L. analyzed clinical data. M.I., A.Mu., S.C.G., and S.S.S. designed, fabricated, and characterized the devices. M.I. and M.T.P.D. performed in vitro experiments. A.L.M., F.W.L., and Q.L. performed in vivo experiments and flow cytometry. J.J.J. and K.R.R. tested the authenticity of and assisted in culturing the cancer cell lines and provided insight into clinical relevance of the findings. F.W.L. and S.S.S. supervised and directed this project. F.W.L., S.S.S., and M.I. analyzed data and wrote the manuscript. All authors critically reviewed and commented on the manuscript.

## Competing interests

S.C.G. and S.S.S. are inventors listed on U.S. Patent #9,789,235 entitled 'Separation and concentration of particles', which describes the 'controlled incremental filtration' technology. They are also co-founders of Halcyon Biomedical Incorporated, a company that would potentially benefit from the commercialization of this patented technology. S.S.S. has received research funding from Halcyon Biomedical Incorporated. All other authors declare no competing interests.
