## [Transparent Peer Review file · Nature Communications]

Ultra-low extracorporeal volume microfluidic leukapheresis is safe and effective in a rat model

Corresponding Author: Dr Fong Lam

Version 0:

Reviewer comments:

Reviewer #1

(Remarks to the Author)

This paper presents a very interesting and practical method for removal of WBCs through low flow rate leukapheresis that can be applied to children with leukemia. The Controlled Incremental Infiltration (CIF) microfluidic platform potentially overcomes challenges in centrifugation based leukapheresis through ultra-low ECV, high collection efficiency for leukemic blasts, minimal PLT loss and the ability to operate effectively at lower flow rates. The study is applied to rats in-vivo and demonstrates equal or higher WBC reduction efficiency. However, the platform still needs further investigation for application in human child. Before publication, I suggest authors to address the following comments.

1. Authors mention that "When designing a CIF element, the widths of the retentate and filtrate channels are gradually adjusted (using a previously described iterative framework) to accommodate the outflow of fluid from the retentate channel and to keep the width of the extracted flow lamina constant along the length of the device. The amount of fluid extracted through each gap and the total number of gaps in the design determine how much of the initial sample volume is ultimately collected as filtrate relative to retentate (i.e., device flow ratio)". Although the principles of iterative framework and choosing the number of gaps are presented previously, authors need to bring the calculations and the numbers they used for this particular study. The authors should bring the geometry of their entire microfluidic device in the supplementary document.
2. It's unclear What is the use of their manifold layer. The retentate could be collected from one port at the main layer without the need for a manifold layer.
3. What does the flow rate of 1.2mL/min mean in the context of leukapheresis? Is it too low even for a child as patient? Will this significantly increase the procedure time?
4. Authors state that "We generated four different designs of the same overall architecture (Figs. 2a-c) but with different flow fractions extracted through the filtration gaps". However, Fig 2a-c do not show those four different designs! Please bring all those four designs and explain why they show different flow fractions through the filtration gaps.
5. "The animals were 200 anticoagulated with heparin", should not it be tubes to be anticoagulated and not the animals?
6. Explanations of figure 5 on page 7 of the pdf has question marks next to the references, please amend this.
7. "During our animal experiments, the flow of blood through the CIF device was driven by the arteriovenous pressure difference alone." Definitely, using no pressure system makes this approach ideal, However, what is the flow rate this pressure gives for their CIF system? How the flow rate and pressure fluctuate during their experiment?
8. Can authors provide more insight into how the data they collected during their experiment on rats can be used in the human?

Reviewer #2

(Remarks to the Author)

This work reports microfluidics-based leukapheresis technique, which has great potential as an alternative to centrifugation-based leukapheresis. The microfluidic device is based on controlled incremental filtration developed at the authors' group, and the authors tested and characterise the separation performance of large leukocyte and spiked leukemic blasts from undiluted whole blood using the device. Besides, the device was tested for small animals - Sprague-Dawley rats for leukapheresis, and the plasma biomarkers and end-organ histology indicate the general safety of the device for leukapheresis. Although the manuscript is well-organised and clearly written, I have some concerns about the novelty and applicability of the device to human patients.

This work is an incremental work based on the authors' previous development on controlled incremental filtration in Refs [27, 28, 29, 30, 32]. Especially, in ref[30], a very similar work has been reported, where the authors reported a high-throughput microfluidic device for centrifugation-free, low extracorporeal volume leukapheresis using the same principle of controlled incremental filtration. In my opinion, the novelty of the technique is not significant enough, and the main improvement relies on the tests of the small animal - Sprague-Dawley rats for leukapheresis. One major motivation of the technique is for therapeutic leukapheresis in pediatric patients. However, the current manuscript does not sufficiently achieve this objective. The techniques have not been tested either pediatric patients or large animal with similar sizes.

In the controlled incremental filtration, the separation of cells of different sizes is based on the flow streamline division at the gap region between the middle channel and adjacent side channels. In this paper, the filtration gap between two arrays of identical is (~19 μm -wide). Although this size is much bigger than single blood cells (red blood cells, platelets and most white blood cells), the filtration gap could be easily blocked by cell aggregation, especially for platelets, which are prone to be activated by fluid shear stress. This brings significant potential risks to the proper functioning of the device. This should bring the authors' attention to address this issue.

Although the plasma biomarkers and end-organ histology indicate no adverse effects compared to sham control, there is no data on the long-term effects of the microfluidic-based leukapheresis. This is especially important when translating the techniques for the treatment of human patients.

In [30], you achieve a high throughput rate of 10mL/min. In this work, you shrink down the throughput to 1.2 mL/min. I can't see a significant challenge in shrinking down the device's throughput in the manuscript. If there is, you may need to elaborate on the relevant technical challenge in more detail to highlight the technical novelty.

Finally, I am concerned about the clogging issue due to the complex blood environment. When the leukapheresis process in the in-vivo experiment lasts for a long time (~3 hours), it could increase the risk of clogging. Therefore, is there a special surface treatment for the PDMS device to avoid it? Or have any other ways been applied to prevent this from happening?

Reviewer #3

(Remarks to the Author)

Version 1:

Reviewer comments:

Reviewer #1

(Remarks to the Author)

The authors have addressed the comments very well. The manuscript reads publishable.

Reviewer #2

(Remarks to the Author)

I appreciate the authors' response to the comments and the efforts to address them. In the revised manuscript, the authors have significantly improved the sections on the novelty, technical advancements, clinical relevance for low extracorporeal volume leukapheresis in the device. The revised manuscript is much more transparent regarding technical innovation, clinical significance and clogging issues. Developing microfluidic devices capable of efficiently separating leukocytes from undiluted whole blood for extended recirculation brings many opportunities not only for leukapheresis but also for blood-based processing and dialysis therapy. After careful evaluation of the revised manuscript, I am happy to recommend the paper for acceptance for publication.

Reviewer #3

(Remarks to the Author)

Dear Editor,

We thank the Reviewers for their thoughtful review of our manuscript. In response to their insightful comments, we made substantial revisions to improve the strength of our manuscript (the changes in the revised manuscript are highlighted in yellow). We also performed new experiments using multiplexed devices to show the scalability of the CIF system. We added a new **figure (Fig. 3)** to the main body and additional figures and tables to the **supplemental file**. Finally, we added Data Availability and Acknowledgement statements. Our point-by-point responses to the Reviewers' comments are given below:

Reviewer #1 (Remarks to the Author):

Specific Comment 1.1: *1. Authors mention that “When designing a CIF element, the widths of the retentate and filtrate channels are gradually adjusted (using a previously described iterative framework) to accommodate the outflow of fluid from the retentate channel and to keep the width of the extracted flow lamina constant along the length of the device. The amount of fluid extracted through each gap and the total number of gaps in the design determine how much of the initial sample volume is ultimately collected as filtrate relative to retentate (i.e., device flow ratio)”. Although the principles of iterative framework and choosing the number of gaps are presented previously, authors need to bring the calculations and the numbers they used for this particular study. The authors should bring the geometry of their entire microfluidic device in the supplementary document.*

Reply: As requested, we added a new supplementary figure (**Supplementary Fig. 1**) which shows the overall geometry of the entire microfluidic device and includes a comprehensive table summarizing the parameters used to build each device design.

Specific Comment 1.2: *2. It's unclear What is the use of their manifold layer. The retentate could be collected from one port at the main layer without the need for a manifold layer.*

Reply: Each device consisted of eight separate CIF elements. In each CIF element, the outlet of the retentate channel was surrounded by filtrate channels (see **Fig. 2a** and **Supplementary Fig. 1a-b**), thus preventing us from connecting the retentate outlets within the main/device layer. To avoid using eight individual collection tubes for each retentate channel, we designed a manifold/top layer to collect the outputs from the eight retentate channels and direct the combined retentate output into a single outlet (**Figs. 2a-c**).

We revised **Results** (page 5) and **Methods** (page 11) to further clarify the need for the manifold layer.

Specific Comment 1.3: *3. What does the flow rate of 1.2mL/min mean in the context of leukapheresis? Is it too low even for a child as patient? Will this significantly increase the procedure time?*

Reply: Combining multiple CIF elements arranged in parallel into a single device is the most straightforward way to increase the throughput without sacrificing separation efficiency. Each CIF element we designed for this study could operate at up to ~0.15 mL/min before elastic PDMS deformations began to diminish separation efficiency appreciably. We chose to multiplex eight of such CIF elements in parallel to align with the throughput requirements of our recirculation experiments in the rat animal model. These eight-element CIF devices operated at up to 1.2 mL/min when driven by a syringe pump *in vitro*, and at ~0.45 mL/min when driven by the arteriovenous pressure difference of the animal *in vivo*. Limiting the degree of multiplexing enabled us to achieve

comparable processing rates between rats (1.1 ± 0.1 of animal's eTBV per hour) and our pediatric patient cohort (0.8 [0.6-1.0] of patient's eTBV per hour).

For this revision, we performed additional experiments to demonstrate that through additional multiplexing the flow rate can be further increased without a significant loss of separation performance (see new **Figure 3**). We compared the performance of a single eight-element CIF device (1.2 mL/min) with that of two (2.4 mL/min), four (4.8 mL/min), and eight (9.6 mL/min) such CIF devices multiplexed in parallel (**Fig. 3a-b**). We achieved a similar level of cell separation performance as we increased the number of CIF elements. These data suggest that the flow rate of CIF-based leukapheresis can be scaled up to meet the throughput requirements of a particular application.

We added the following paragraph to the **Discussion** (page 9) to discuss the meaning of these findings in the context of leukapheresis: "The ECV of the microfluidic leukapheresis circuit in our animal experiments was 0.74 mL, of which only a small fraction was due to the dead volume of the CIF device (<0.1 mL). In our parallelized eight-device version (64 CIF elements, Fig. 3) the device and the tubing had a dead volume of ~4.4 mL which is about 1/70th of the ~300 mL ECV of a typical centrifugation-based leukapheresis circuit currently used in clinical practice. Such a dramatic reduction of ECV would be particularly important in leukemic infants with hyperleukocytosis who are often too small to perform centrifugation-based leukapheresis safely¹⁴. In our clinical cohort, the smallest patient receiving leukapheresis was 9 kg (7.2 months old). Because of their small size, they required mechanical ventilation for placement of the large bore catheter, circuit blood priming, and became hypotensive during leukapheresis. A smaller device with similar efficacy, like CIF-based leukapheresis, could potentially mitigate most of these adverse effects or need for additional resources. For example, our multiplexed eight-device system ran at 9.6 mL/min (576 mL/hr). For a 9-kg patient, that flow rate would result in a processing rate of 0.91 eTBV/hr which is on par with our current clinical practice (0.8 [0.6-1.0] eTBV/hr, Fig. 1I). Importantly, ECV of our eight-device multiplex would represent only 0.7% of the patient's eTBV, which is drastically lower than for centrifugation-based leukapheresis (~48% of the patient's eTBV)."

Specific Comment 1.4: *Authors state that "We generated four different designs of the same overall architecture (Figs. 2a-c) but with different flow fractions extracted through the filtration gaps". However, Fig 2a-c do not show those four different designs! Please bring all those four designs and explain why they show different flow fractions through the filtration gaps.*

Reply: As stated in our reply to **Specific Comment 1.1** above, we added a new supplementary figure to better illustrate the overall architecture of the CIF devices and provide the parameters we used to build the four designs (see **Supplementary Fig. 1**). The initial value of the extracted flow fraction (f_{gap}^*) was one of the parameters we set *a priori* when calculating each design. For a given middle channel width, the value of f_{gap}^* determines the width of the extracted flow lamina and therefore the cell size cutoff for the design. The larger the value of f_{gap}^* , the larger the cells that can be pulled by the filtrate into the side channels of the device. Please see **Methods** (pages 11-12) for additional details on CIF device development.

Specific Comment 1.5: *"The animals were 200 anticoagulated with heparin", should not it be tubes to be anticoagulated and not the animals?*

Reply: When performing animal experiments, all parts of the extracorporeal circuit (catheters, microfluidic devices, connecting tubing, etc.) were anticoagulated with 10 U/mL heparinized saline. The animals were given intravenous heparin (500 U/kg) for

anticoagulation prior to connecting a microfluidic device (CIF group) or a resistance bridge (sham group).

We revised **Results** (page 6) and **Methods** (pages 11 and 13) to further clarify our anticoagulation approach.

Please also see our reply to **Specific Comment 2.5** below.

Specific Comment 1.6: *Explanations of figure 5 on page 7 of the pdf has question marks next to the references, please amend this.*

Reply: We made the correction, as requested.

Specific Comment 1.7: *“During our animal experiments, the flow of blood through the CIF device was driven by the arteriovenous pressure difference alone.” Definitely, using no pressure system makes this approach ideal, However, what is the flow rate this pressure gives for their CIF system? How the flow rate and pressure fluctuate during their experiment?*

Reply: We included a plot of the flow rate over time of our recirculation experiments *in vivo* (**Figure 4c**), as requested. The average flow rate for the CIF device was 0.448 ± 0.032 mL/min (this value was added to the **Results**). The flow rates in both the CIF and sham groups remained consistent throughout the experiments. We updated **Fig. 4c** to now show both the continuous mean arterial pressure (MAP) and mean total flow rate (30-minute epochs).

The **Fig. 4c** caption now reads: “(c) Although animals in the CIF group had an overall lower mean arterial blood pressure (MAP) and higher flow throughout the procedure than those in the sham group, there were no differences in the trend (slope) of MAP or flow between the groups. The flow rates are expressed at 30-minute epochs.

**** $p < 0.001$ between groups for the y-intercepts only.”

Additionally, we added the following text in the revised **Results** (page 6): “Likewise, the mean total flow rate remained stable throughout the experimental duration (**Fig. 4c**). While the overall mean flow rate was higher in the CIF compared to the sham group (0.448 ± 0.032 vs. 0.319 ± 0.043 mL/min, respectively; $p < 0.05$), the number of eTBV processed was similar between the two groups (**Fig. 4d**).”

Finally, we added the following explanation in the revised **Methods** (page 14): “The mean flow rate was expressed as 30-minute epochs (e.g., 0-30 minutes, 30-60 minutes, 60-90 minutes, etc.) by adding the mean filtrate flow rate to the calculated weighted mean retentate flow rate (volume collected divided by time of collection) for each 30-minute epoch.”

Specific Comment 1.8: *8. Can authors provide more insight into how the data they collected during their experiment on rats can be used in the human?*

Reply: As requested, we included the following explanation of the usefulness of our rat data in the context of human leukapheresis in the revised **Discussion** (page 8): “Our data in rats show the potential benefits of CIF-enabled microfluidic leukapheresis over the currently used centrifugation-based method. First, our rat model shows that microfluidic leukapheresis can occur without the need for a driving pump to achieve a similar amount of blood processing (~1 TBV/hr) and leukocyte removal (~46%) as performed in children at our institution. This could greatly simplify leukapheresis by reducing the need for the large bore venous catheters typically used to perform centrifugation-based leukapheresis. Leukapheresis could potentially be performed using

peripheral artery to vein connections that typically require less sedation to place. Second, this amount of blood processing is possible using a system with minimal ECV. Third, microfluidic leukapheresis did not incur organ injury in our rat model. The slight rise in creatinine and blood urea nitrogen occurred in both the CIF and sham groups and likely represented repeated blood collection for serial CBC. Finally, microfluidic leukapheresis had minimal effect on PLT activation or inflammation. Taken together, these data suggest that CIF-enabled microfluidic leukapheresis may be safe to use in humans in lieu of other, more invasive, methods. However, longitudinal follow-up studies (weeks/months post-treatment) in large animal models are necessary to ensure a lack of negative long-term sequelae from CIF-based leukapheresis.”

In the revised **Discussion** (page 9), we also expanded on how – with additional multiplexing – the CIF device can be used in pediatric patients (see our reply to **Specific Comment 1.3** above).

Reviewer #2 (Remarks to the Author):

Specific Comment 2.1: *This work is an incremental work based on the authors' previous development on controlled incremental filtration in Refs [27, 28, 29, 30, 32]. Especially, in ref[30], a very similar work has been reported, where the authors reported a high-throughput microfluidic device for centrifugation-free, low extracorporeal volume leukapheresis using the same principle of controlled incremental filtration. In my opinion, the novelty of the technique is not significant enough, and the main improvement relies on the tests of the small animal - Sprague-Dawley rats for leukapheresis. One major motivation of the technique is for therapeutic leukapheresis in pediatric patients. However, the current manuscript does not sufficiently achieve this objective. The techniques have not been tested either pediatric patients or large animal with similar sizes.*

Reply: Our study represents two significant firsts: (a) This is the first demonstration of safely removing leukocytes from recirculating blood using a microfluidic device in a live animal model. (b) Secondly, it achieves highly efficient collection of leukocytes and leukemic blasts directly from undiluted whole blood samples at flow rates that are clinically relevant for applications like leukapheresis. The separation of leukocytes from undiluted whole blood with high efficiency and throughput continuously over extended periods of time without clogging and activating/damaging the cells has been a major challenge for microfluidic cell separation. Our study solved this problem, paving the way towards further testing in large animal models and human volunteers to ultimately enable safe leukapheresis for neonates and infants, the most vulnerable patients who are currently underserved.

We believe that the novelty of our work lies in combining the latest CIF design innovations to create a CIF device with an entirely new function required for a highly significant medical application (ultra-low ECV leukapheresis). While it is certainly true that we described the basic operating principle of 'controlled incremental filtration' (CIF) as early as in our Lab Chip 2014 article (**ref. [27]**), it has taken a significant amount of research and development to make the approach truly versatile and practical. The culmination of this innovative work is the CIF device described in this manuscript, which is capable of separating leukocytes from undiluted whole blood – a functionality that until now has not been demonstrated by any other microfluidic technology, including our own previous work (i.e., **ref. [30]**).

In this study we made significant design improvements (relative to **ref. [30]**) to develop a CIF device uniquely capable of processing undiluted whole blood: (i) re-invented the way the widths of the central (retentate) and side (filtrate) channels change along the length of

the device (see new **Supplementary Fig. 1**) to account for the changes in local blood viscosity due to increasing leukocyte concentration and hematocrit variations, *(ii)* eliminated high-shear design elements (such as the pre-element filters) to minimize platelet activation which may cause aggregate formation and consequent plugging of the device, *(iii)* optimized the fraction of flow extracted through the gaps (f_{gap}^*) to remove large leukocytes and leukemic blasts while minimizing hematocrit differences between the retentate and filtrate, and *(iv)* increased the ratio of filtrate to retentate (flow ratio) to significantly reduce the loss of RBCs and platelets.

For this revision, we performed additional experiments to demonstrate that through multiplexing the flow rate can be increased to ~10 mL/min without any loss in separation efficiency (see **Fig. 3, Supplementary Fig. 4**). Such a throughput is already sufficient to perform ultra-low ECV leukapheresis in pediatric patients (see our reply to **Specific Comment 1.3** and **1.8**). We revised the **Introduction** (page 3) to further emphasize that a novel device had to be developed to enable the desired functionality (processing of undiluted whole blood). Finally, we added a new **Supplementary Figure 1** to illustrate the geometry of the novel device developed in this study.

Specific Comment 2.2: *In the controlled incremental filtration, the separation of cells of different sizes is based on the flow streamline division at the gap region between the middle channel and adjacent side channels. In this paper, the filtration gap between two arrays of identical is (~19 μm -wide). Although this size is much bigger than single blood cells (red blood cells, platelets and most white blood cells), the filtration gap could be easily blocked by cell aggregation, especially for platelets, which are prone to be activated by fluid shear stress. This brings significant potential risks to the proper functioning of the device. This should bring the authors' attention to address this issue.*

Reply: We agree with **Reviewer #2** that clogging by cell aggregates is a major concern for virtually any microfluidic cell separation technique, and activation of platelets by shear can certainly lead to the formation of cell aggregates. In our previous studies, we purposefully optimized the overall CIF design to minimize shear in the separation channels (see **ref. [28]**), and we found that CIF devices caused a minimal increase (~2 percentage points) in platelet activation (% of CD62P+ platelets) relative to the initial whole blood (WB) sample (~3% activation in WB vs. ~5% after CIF) which was ~4-fold lower than after centrifugation (an increase of 9 percentage points to ~12%) (see **ref. [31]**). In this study, we further improved the CIF device design by identifying and eliminating any additional design elements that could expose platelets to excessive shear (see our reply to **Specific Comment 2.1**).

In the revised manuscript, we included an estimation of the shear experienced by the cells in the CIF device (see new **Supplementary Note**). Within the filtration gaps, the shear rate and stress were quite low and well below the values required for platelet activation reported in the literature. At the start of the CIF element, where the shear is expected to be the highest, the values for shear rate and stress were still around the physiological range observed in micro arterioles. As described in our previous studies (e.g., see **ref. [30]**), platelets that become activated in the CIF device tend to bind to leukocytes and form platelet-leukocyte aggregates. However, in this study, we found no difference in platelet-leukocyte aggregate formation between the CIF or the sham groups, nor the retentate containing leukocytes removed from the animal during the procedures (see **Fig. 6dd**). This result suggests that CIF devices cause only minimal platelet activation.

As **Reviewer #2** correctly states, CIF works by extracting a relatively small fraction of the central channel flow through each filtration gap. The width of this extracted flow lamina (not the width of the gaps) determines the maximum size of the particles/cells that can be carried by the filtrate through the gaps into the side channels. Particles/cells that are larger than the extracted flow lamina width remain in the retentate. Therefore, any cell aggregates that may form in the CIF device (or enter the device from the inlet) are typically too large to be pulled by the filtrate through the gaps and instead remain in the central channel to be extracted with the retentate. This inherent property of the CIF separation approach significantly enhances the safety of CIF-based microfluidic leukapheresis by effectively preventing cell aggregates from returning to the body with the filtrate. It is in part because of this serendipitous feature we were able to operate our CIF devices without any clogging, any decrease in flow rate or separation performance, or any adverse effects on the animal indicative of micro-embolism for >3 hours in the rat leukapheresis model (see **Fig. 4c**, **Supplementary Fig. 5c-d**, **Fig. 5**).

Finally, we included a new supplementary video (**Supplementary Video 2**) showing the typical dynamics of blood flow within the device that is free of cell aggregates and clogging.

Specific Comment 2.3: *Although the plasma biomarkers and end-organ histology indicate no adverse effects compared to sham control, there is no data on the long-term effects of the microfluidic-based leukapheresis. This is especially important when translating the techniques for the treatment of human patients.*

Reply: We appreciate **Reviewer #2's** point regarding long-term effects of microfluidic-based leukapheresis. While our current study demonstrates no immediate adverse effects as evidenced by plasma biomarkers and end-organ histology, we acknowledge the need for long-term safety data, particularly for clinical translation.

In this study, the CIF device prototypes were fabricated from PDMS, which is a platinum-cured silicone. Therefore, we anticipate that long-term effects of exposure to a PDMS device would be similar to other platinum-cured silicones currently being used in many infusion applications (such as components of implantable central venous port devices and extracorporeal membrane oxygenation circuits) with excellent safety profiles. However, the eventual commercial CIF devices will likely be made from medical-grade thermoplastics (not from PDMS) and therefore their long-term safety will need to be evaluated *de novo*.

At the same time, there are very few studies examining the long-term physiological effects of a leukapheresis procedure itself in either animal or human subjects. For example, it would be particularly interesting to investigate the effect on bone marrow from multiple repeated procedures. Our scalable, ultra-low ECV microfluidic leukapheresis devices could enable such mechanistic studies in both small (mouse, rat) and large (pig) animal models for the first time.

We revised the **Discussion** section (page 8) to further highlight the need for investigating long-term effects in future studies: "... However, longitudinal follow-up studies (weeks/months post-treatment) in large animal models are necessary to ensure a lack of negative long-term sequelae from CIF-based leukapheresis."

Specific Comment 2.4: *In [30], you achieve a high throughput rate of 10mL/min. In this work, you shrink down the throughput to 1.2 mL/min. I can't see a significant challenge in shrinking down the device's throughput in the manuscript. If there is, you may need to elaborate on the relevant technical challenge in more detail to highlight the technical novelty.*

Reply: In [30], each device comprised 48 CIF elements multiplexed in parallel. While it is true that these devices operated at 10 mL/min, to achieve optimal separation efficiency the blood had to be diluted down to 10% hematocrit, and the separation performance was severely diminished at higher hematocrits. Therefore, making CIF devices capable of separating leukocytes from **undiluted whole blood** efficiently during extended recirculation was the primary technical challenge of this study. Overcoming this challenge represents a transformative leap forward toward enabling ultra-low ECV leukapheresis for pediatric patients (also see reply to **Specific Comment 2.1**).

As mentioned above (see **Specific Comment 1.3**), we have performed new experiments demonstrating that through multiplexing the flow rate for our new CIF device design can be increased to ~10 mL/min without any loss of separation performance, even for **undiluted whole blood** (see new **Fig. 3**).

Specific Comment 2.5: *Finally, I am concerned about the clogging issue due to the complex blood environment. When the leukapheresis process in the in-vivo experiment lasts for a long time (~3 hours), it could increase the risk of clogging. Therefore, is there a special surface treatment for the PDMS device to avoid it? Or have any other ways been applied to prevent this from happening?*

Reply: We had the same concern, and therefore we worked to mitigate clogging in our device by eliminating design elements that could cause excessive shear (see our reply to **Specific Comment 2.2** above) and by applying a special treatment to the surface of PDMS to reduce cell adhesion (see description in the original **Methods**, page 12).

In the revision, we included a new **Supplementary Figure 7** to better illustrate the difference between a treated device and an untreated device. The treated device was incubated with 1% (w/v) mPEG-Silane (MW 5000) solution, followed by a flush and incubation with 1% (w/v) BSA solution overnight. Such treatment virtually eliminated device clogging (see **Supplementary Fig. 7**).

When performing animal experiments, thus treated devices were additionally flushed and incubated (for at least 30 minutes) with heparinized saline (10 U/mL) before each run. Also, prior to connecting the CIF device (or sham bridge) to the animals, we primed the extracorporeal circuit with heparinized saline (10 U/mL) to remove air. Finally, the animals were anticoagulated with 500 U/kg of heparin. We anticipated that device clogging may reduce the flow rate and/or reduce the separation efficiency. However, in our experiments the flow rates did not change throughout the whole duration of the 3-hour recirculation procedures (**Fig. 4c**) and the separation performance of the CIF devices actually increased over time (**Supplementary Fig. 5c-d**). These data suggest that our multi-pronged approach to mitigating device clogging was successful. We revised the **Methods** (page 12) to expand the description of our clogging mitigation approach.

Reviewer #3 (Remarks to the Author):

Specific Comment 3.1: *I co-reviewed this manuscript with one of the reviewers who provided the listed reports. This is part of the Nature Communications initiative to facilitate training in peer review and to provide appropriate recognition for Early Career Researchers who co-review manuscripts.*

Reply: We thank **Reviewer #3** for taking the time to review our manuscript.

In addition to the changes introduced in response to the comments of the Reviewers (above) we made several minor corrections throughout the revised manuscript to improve its readability.